

# Quantifying the effect of forests on occurrence frequency and intensity of rockfalls

Christine Moos[1], Luuk Dorren[1], Markus Stoffel[2,3,4]

[1]Berne University of Applied Sciences, School of Agricultural, Forest and Food Science HAFL, Länggasse 85, CH-3052 Zollikofen, Switzerland

[2] Climatic Change and Climate Impacts, Institute for Environmental Science, University of Geneva, 66 Bvd Carl Vogt, CH-1205 Geneva, Switzerland

[3] Department of Earth Sciences, University of Geneva, rue des Maraîchers 13, CH-1205 Geneva, Switzerland

[4] Dendrolab.ch, Institute of Geological Sciences, University of Bern, Baltzerstrasse 1+3, CH-3012, Bern, Switzerland

*Correspondence to*: Christine Moos (christine.moos@bfh.ch)

**Abstract.** Forests serve as a natural means of protection against rockfall. Due to their barrier effect, they reduce the intensity
and the propagation probability of falling rocks and thus the occurrence frequency of an event for a given element at risk. However, despite established knowledge on the protective effect of forests, they are mostly neglected in quantitative rockfall risk analysis. Their inclusion in quantitative rockfall risk assessment would, however, be necessary to express their efficiency in monetary terms and to allow comparison of forests with other protection measures. The goal of this study is to quantify the effect of forests on the occurrence frequency and intensity of rockfalls. We therefore defined a rockfall onset
frequency based on a power-law magnitude-frequency distribution, which then served as input for the simulation of rockfall events on a virtual slope and over a period of 1000 yrs. Simulations were run for different forest and non-forest scenarios under varying forest stand and terrain conditions. We determined rockfall frequencies and intensities at five different horizontal distances from the release area. Based on two multivariate statistical prediction models, we investigated which of the terrain and forest characteristics are predominantly driving the role of forest in reducing rockfall occurrence frequency
and intensity and whether they are able to predict the effect of forest on rockfall occurrence and hence rockfall risk. The rockfall occurrence frequency below forested slopes is reduced between approximately 10 and 90 % as compared to non-forested slope conditions; whereas rockfall intensity is reduced by 10 to 70 %. This reduction increases with increasing slope length and decreases with decreasing tree density, tree diameter and increasing rock volume as well as in case of clustered or gappy forest structures. The statistical prediction models reveal that the cumulative basal area, rock volume and horizontal
forest structure represent key variables for the prediction of the reducing effect of forests. In order to validate these results, models have to be tested on real slopes with a wide variation of terrain and forest conditions.



## 1.    Introduction

Rockfall is a widespread and frequent natural hazard occurring below steep rocky cliffs. The occurrence of rockfall often threatens infrastructures, transportation corridors, and human life. We here define it as a fragment of rock detaching from a release area and proceeding downslope by bouncing, falling, or rolling (Whittow, 1984). Different protective measures are typically implemented in order to reduce risks in rockfall prone areas. These include structural protection measures, land-use planning, early-warning systems or biological measures, nowadays referred to as nature-based or eco- solutions (Agliardi and Crosta, 2003; Corominas et al., 2005; Sättele et al., 2016; Renaud et al., 2013). With regards to rockfall, a well-known biological measure is the protection forest. Such forests can serve as a natural means of protection against rockfall due to their barrier effect. Forests influence rockfall risk by (i) reducing the intensity of falling rocks after collisions with tree stems and by (ii) reducing the propagation probability and thus the occurrence frequency of an event at a given element at risk (Wasser and Perren, 2014).

In order to appropriately account for the positive effects of protective measures on rockfall risk and associated uncertainties, their design should be based on a quantitative risk analysis (Corominas et al., 2005; Straub and Schubert, 2008; Peila and Guardini, 2008). In doing so, the protective effect of the measure can be expressed in monetary terms, thereby allowing to evaluate its efficiency in a cost-benefit analysis (Agliardi et al., 2009). In the case of protection forests, quantitative, risk-based approaches have been only rarely applied in the past. Despite the advanced knowledge on the protective effect of forests and its maintenance (Dorren et al., 2007; Bigot et al., 2009), open questions remain on how protection forests can be quantitatively integrated into rockfall risk analyses (Masuya et al., 2009; Trappmann et al., 2014). Currently, the effect of forests is mostly neglected or only qualitatively assessed in hazard and risk analyses.

The quantification of the influence of forests on rockfall occurrence frequency is particularly demanding, especially if one aims at evaluating the effect of forests at the level of the element at risk. The occurrence frequency of a rockfall event is usually described by the annual exceedance frequency of its magnitude (expressed as the rockfall volume) or intensity (expressed as the kinetic energy of the blocks), assuming that rockfall occurrence follows a Poisson distribution (Corominas et al., 2013). Depending on the data availability and site characteristics, the annual exceedance frequency can be estimated by different approaches including the analysis of historical datasets (Hantz et al., 2003; Hungr et al., 1999; Guzzetti et al., 2003), magnitude-frequency relationships based on power laws (e.g. Agliardi et al., 2009; Lari et al., 2014; Dussauge-Peisser et al., 2002), empirical models describing rockfall frequency as a function of topographic or geological parameters (e.g. Budetta, 2004; Lan et al., 2010), or expert opinion (e.g. Romana et al., 2003). Furthermore, several techniques exist based on which the depositional ages of rocks can be reconstructed in absolute terms (e.g. Lang et al., 1999; McCarroll et al., 2001).



Dendrogeomorphology (Stoffel and Corona, 2014) represents one such approach and has proven to be a reliable method to estimate past rockfall frequencies through coupling the number of rockfall impacts with tree age (Moya et al., 2010; Corona et al., 2013; Trappmann et al., 2014; Perret et al., 2006). However, in most cases, reliable data is scarce and estimation of robust occurrence frequencies remains difficult (Hantz et al., 2003; Lari et al., 2014; Straub, 2005). Therefore, practitioners usually assume scenarios of pre-defined return periods and corresponding rock volumes (e.g. Borter et al., 1999). Such scenarios are typically derived for the current (e.g. forested) situation, but are also applied to reference (e.g. non-forested) situations (Jahn, 1988). At the same time, however, the barrier effect of forests is expected to decrease the occurrence frequency of rockfall at the location of the element at risk. Consequently, scenarios derived with the practitioner's approach may not necessarily be valid for the reference situation and might thus result in biased risk estimations.

Forests do not only reduce the occurrence frequency of rockfall events, but also reduce their intensity by stopping rocks completely and/or by absorbing (part of) their energy (Lundström et al., 2009). In this sense, the intensity of an event refers to the kinetic energy which is released by the rock at impact with the element at risk (Jaboyedoff et al., 2005; Abbruzzese et al., 2009; Lari et al., 2014).

The effect of forest on the occurrence frequency and the intensity is also expected to depend on the structure of a forest stand. Furthermore, the capacity of a tree to absorb energy will vary between species and will depend on its diameter at breast height (DBH) (Dorren et al., 2006). At the stand level, high stem densities are considered to stop falling rocks more effectively because of an enhanced impact probability (Dorren and Berger, 2005; Wehrli et al., 2006). The three-dimensional, probabilistic-deterministic rockfall simulation model RockyFor3D (Dorren, 2015) accounts for these forest effects. It integrates trees spatially explicitly and calculates the energy loss due to impacts against single trees as a function of tree species, DBH, impact height and the horizontal position of the hit (Dorren et al., 2006).

The goal of this study is to quantify the effect of forests on the occurrence frequency and intensity of rockfall by using multiple series of rockfall simulations. In this paper, we define a rockfall onset probability based on a power-law magnitude-frequency distribution, which then serves as input for the simulation of rockfall events on a virtual slope and over a period of 1000 yrs. Simulations were run for different forest and non-forest scenarios under varying forest stand and terrain conditions. These simulations do not only (i) provide input data for the determination of rockfall occurrence frequencies and intensities at five different distances from the release area, but also (ii) yield information how specific forest and terrain characteristics control rockfall occurrence frequency and intensity along a slope. The second question was addressed using multivariate statistical prediction models. Based on these approaches, we then investigate (iii) how rockfall occurrence frequency and intensity differ at a given location with an element at risk on forested and non-forested slopes; (iv) what terrain and forest characteristics are predominantly driving the role of forest in reducing rockfall occurrence frequency and intensity, and (v) whether multivariate statistical models fitted with these terrain and forest characteristics can indeed predict the effect of forest on rockfall occurrence and hence rockfall risk.



## 2.    Material and methods

As this study aimed at an assessment of rockfall in forests under controlled conditions, it was preferred to run simulations on a virtual slope. We designed a slope raster with a resolution of 2 m, a horizontal width of 478 m and a horizontal length of

574 m. The virtual slope has a concave shape and slope angles which are increasing linearly from 20 to 40° from the slope bottom to the release area of rockfall, therefore resulting in a height difference of 328 m. The slope angle of each cell was randomly varied with ± 10 %. The release area of rockfall is rectangular and has a horizontal length of 100 m and a width of 300 m. The release area was placed in the central part of the uppermost slope area and defined as a series of rockfall cliffs. Initial fall height of rocks was defined as 10 m above ground. At the bottom part of the slope, we designed a horizontal road

with a width of 10 m and we further added five virtual evaluation lines located at horizontal distances of 0, 150, 300, 450, and 530 m from the downslope side of the release area to the bottom of the slope. These lines allow a systematic assessment of changes in rockfall occurrence frequency and intensity with increasing distance from the release area of rockfalls (Fig. 1). The lines were defined based on equal height differences between them. Noteworthy, the evaluation line at a slope length of 450 m is identical to the level of the road that was built into the model.

### 2.1.    Rockfall simulation model

We used the model RockyFor3D for the rockfall simulations, which is a probabilistic process-based rockfall trajectory model simulating trajectories of falling rocks using 3D vectors (Dorren, 2015). RockyFor3D was developed on the basis of real-size rockfall experiments in the field and uses raster maps describing topography (Digital Elevation Model, DEM), rockfall

source cells, the elasticity of the surface material, slope surface roughness, the number of trees per cell, DBH of trees in each cell and tree species per cell as input data (Dorren et al., 2004; Dorren et al., 2006). For each rockfall source cell, the trajectories of a given number of rocks are simulated by considering flying and bouncing. Rolling is simulated with short distance bouncing, similar to the approach of Pfeiffer and Bowen (1989). The trajectory of rockfall is primarily determined by topography. The elasticity of the impacted material is calculated based on the normal coefficient of restitution ($R_n$) which

is predefined by seven different soil types or undergrounds. Surface roughness is represented by a mean obstacle height (MOH) representative for 70, 20 and 10 %, respectively, of each cell (for more details see Dorren, 2015). RockyFor3D explicitly calculates the deviation and energy loss after impacts with trees dependent on tree diameter, impact position, and the kinetic energy of the rock before the impact. Provided that the exact positions of trees within the slope are not known, trees are randomly positioned within each pixel according to the number of trees (i.e. forest stand density) assigned to each

pixel. The main output of RockyFor3D consists of raster cells containing the maximum kinetic energy, the 95 % confidence interval of all maximum kinetic energy values, the maximum bounce height, the number of rocks passed through each cell, the number of deposited rocks, the maximum simulated velocity, the maximum tree impact height and the number of tree impacts per cell (Dorren et al., 2006; Dorren, 2015). We simulated one block per event following a power-law magnitude frequency distribution and did not consider rock fragmentation.



## 2.2.    Onset probability

We assume a power-law distribution for the magnitude-frequency relationship of rocks released from the release area, since power laws have proven to fit the release volume distribution of rockfalls (e.g. Dussauge-Peisser et al., 2002; Malamud et al., 2004) with the general form:

$$F(V_i) = \alpha V_i^{-\beta} \qquad (1)$$

where F($V_i$) is the annual frequency of volume i ($V_i$).

We derived the exponent of the power law (ß) from Carrea et al. (2015) who used Terrestrial Laser Scanning to identify fallen blocks with their respective volumes for the "La Cornalle" site in Switzerland. At this site, the cliff consists of alternate layers of marls and sandstones. The power-law distribution was fitted to volumes ranging from 0.010 m³ to 7.63 m³. For the scope of our study, we considered blocks with volumes between 0.05 m³ and 2.0 m³. These volumes can be potentially hazardous but are still within a range for which forests are assumed to have an effect on rockfall propagation and energy (Stokes, 2006). The constant α of the cumulative power-law distribution was defined as 12 in our study corresponding to a rockfall retreat rate of approximately 0.2 mm/yr for the considered volume range (0.05 m³ and 2.0 m³). This is in the typical range of rockfall retreat rates in alpine regions (Sass and Wollny, 2001; Hoffmann and Schrott, 2002; Moore et al., 2009). The simulations were run over a period of 1000 yr. We calculated the number of expected rockfalls released in 1000 yrs for 20 volume classes (Fig. 2). The source cells were randomly sampled in the release area.

## 2.3.    Forest and terrain scenarios

The soil scenarios (Table 1) considered scree or medium compact soil with small rock fragments (soil type 3) and talus slope or compact soil with large rock fragments (soil type 4), as these are expected to be most frequent, often continuous and with a large spatial distribution. The release area was in all cases defined as soil type 5 (bedrock with thin weathered material or soil cover) and the road was set to soil type 7 (asphalt road). As shown in Table 1, soil roughness was set to 0 m (100 %) in the scenario "zero roughness" and to 0.15 m (10 % of the surface), 0.05 m (20 %) and 0.01 m (70 %) in the scenario "rough", respectively. Definition of the four forest types (Table 2) was based on natural rockfall protection forests as defined from the Swiss National Forest Inventory (Messmer, 2014). The forest types differ with respect to the diameter at breast height (DBH; ranging from 21-40 cm), dominant tree species (deciduous, conifers) and the number of tree stems (with DBH > 12 cm) per hectare (Nha; 200-500 trees ha⁻¹). The forest stands of each forest type were designed for four different horizontal forest structures (Fig. 2) as follows: random tree distribution, clustered tree distribution, random distribution with gaps of 20 x 20 m and random distribution with 3 aisles of 20 m in width.

The combination of the different forest types and structures and terrain scenarios yielded 49 different simulation scenarios. For each scenario, the expected number of blocks of the 20 volume classes was then simulated 100 times to obtain robust results.





## 2.4. Statistical analysis

Simulation results were analysed statistically as follows:

(i)    zonal statistics of rockfall frequencies and energies at the level of the evaluation lines

(ii)   statistical comparison of rockfall frequency and intensity between different scenarios and by fitting power-law based intensity-frequency curves

(iii)  design of multivariate statistical models relating the frequency and the intensity reduction of forests to terrain and forest characteristics

(iv)   assessment of the performance of the statistical models and sensitivity to changes in slope angle

For each volume class and simulation scenario, the number of rocks passing a cell (i.e. number of passages) was summed up for the five evaluation lines (EL) and averaged over the 100 simulations. We calculated the rockfall occurrence frequency (Freq) by dividing the sum of the number of passages (Nrp) by 1000 yrs:

$$Freq = \frac{\sum_{EL} Nrp}{1000\ yrs} \qquad (2)$$

Consequently the return period (RP) of rockfall is its reciprocal value and represented as follows:

$$RP = \frac{1}{Freq} = \frac{1000\ yrs}{\sum_{EL} Nrp} \qquad (3)$$

We calculated an indicator for the reduction in the number of passages by the forest stand ($Nrp_{red}$) in order to evaluate changes in the frequency between forested and non-forested conditions. The indicator $Nrp_{red}$ is defined as the difference between the number of passages without ($Nrp_{nF}$) and with forest ($Nrp_F$), divided by the number of passages without forest (Eq. 4):

$$Nrp_{red} = \frac{Nrp_{nF} - Nrp_F}{Nrp_{nF}} \qquad (4)$$

We then used the 95[th] percentile of the maximum energy (E95 in kJ) as an indicator for rockfall intensity. For each EL, we calculated the mean of E95 and averaged it over the 100 simulations. Similarly to occurrence frequency, we calculated the

intensity reduction offered by forests ($E95_{red}$). This indicator is defined as the difference between E95 without ($E95_{nF}$) and with forest stand ($E95_F$) divided by E95 without forest (Eq. 5):

$$E95_{red} = \frac{E95_{nF} - E95_F}{E95_{nF}} \qquad (5)$$



We further determined intensity-frequency distributions of E95 (intensity) and Nrp (frequency) under different forest and non-forest scenarios and at a slope length of 300 m, to which power-law distributions (Eq. 1) were fitted based on least squares (Draper and Smith, 1998).

To detect possible effects of forest and terrain characteristics on the forest effect, we first assessed whether $Nrp_{red}$ and $E95_{red}$ significantly differ between different forest and terrain scenarios based on the *Wilcoxon rank-sum test*, with a significance threshold of $p \leq 0.05$. Subsequently, we applied regression tree (RT) models (Breiman et al., 1984) and generalized linear models (GLM) (McCullagh and Nelder, 1989) relating $Nrp_{red}$ and $E95_{red}$ to possible explanatory variables.

  RTs are a non-parametric regression approach which recursively partitions the data based on explanatory variables. At each
node, the data is split into two groups using a single predictor (Breiman et al., 1984). The splitting variable is selected aiming at impurity reduction. This means that daughter nodes have to be as homogeneous ("pure") as possible. RTs consider parameter interactions and account for non-linearities (Vorpahl et al., 2012). RT models were fitted using the *rpart* function of the party package in the statistical software R (Ripley et al., 2015).

  We used rock volume, soil type (categorical), soil roughness (categorical), the horizontal forest structure (categorical) and
the cumulative basal area (cbA; Eq. 7) of the forest as potential explanatory variables. The latter is defined as the product of the basal area (bA; $m^2$/ha) for a slope width of 100 m and the forested slope length (fsL; m) from the top of the release area to the respective EL. The basal area (bA) is defined as the area per hectare which is occupied by the cross-section of tree stems (Bitterlich, 1948).

$$cbA\ [m^2 ha^{-1}] = \frac{bA}{100\ m} \times fsL = \frac{\sum_{EL} bA / \sum_{EL} area}{100\ m} \times fsL \qquad (7)$$

  We calculated the Spearman correlation coefficients to check whether the explanatory variables are substantially correlated. The final GLM was determined using a stepwise backward variable selection with the aim to minimize the Akaike Information Criterion (AIC). The quality of the models was examined with goodness-of-fit tests and customary residual
diagnostic plots (Stahel, 2013) indicating that the cumulative basal Area (cbA) should be transformed to the natural logarithm.

  The GLM and RT were fitted with the simulation data of the concave slope. They were subsequently calibrated with a training data set representing 75% of the data. We further applied three times repeated 10-fold cross validation and calculated the average performance across the hold-out predictions with the aim to avoid over-fitting (Kohavi, 1995). The
predictive performance was assessed based on the Root Mean Squared Error normalized with the range of the simulated data (nRMSE).





Furthermore, we tested the statistical prediction models for $Nrp_{red}$ with field data of a study site in the French Alps at which real-size rockfall experiments were conducted on forested and non-forested sites (Dorren et al., 2006). We evaluated $Nrp_{red}$ at a distance of 223 and 324 m from the release point (as measured along the slope).

To assess whether the forest effect on rockfall frequency and intensity depends on the slope angle, we conducted additional simulations for four linearly shaped slopes with varying slope angles (32°, 35°, 38°, 40°) for forest type 1 with random tree distribution, soil type 3 and rough conditions. On these slopes, we tested the multivariate statistical prediction models designed for the concave slope (GLM, RT) and calculated their performance. On the linearly-shaped slopes, evaluation lines were defined with the same distances along the slopes.

## 3. Results

### 3.1. Effect of forest on rockfall occurrence frequency

Forest stands considerably reduce rockfall frequency, with differences in the frequency between the forested and non-forested slope scenarios increasing strongly with increasing slope length. In the case of forest type 1 (*Fagus sylvatica* forest with 460 stems ha$^{-1}$) with randomly distributed trees, the frequency at a distance of 450 m from the release area has been shown to decrease to zero (RP > 1000 yrs) whereas on the non-forested slope, the RP remains at values ranging from 1 to 100 yrs, depending on rock volume (Fig. 4). We also show that with decreasing cbA, the effect of the forest is decreasing ($p < 0.05$; Fig. 6), and the reduction of rockfall is becoming less effective. In a pole-stand *F. sylvatica* forest (forest type 4), by contrast, the return period decreases to values between 30 and 1000 yrs at a slope length of 450 m. In the conifer forest composed of *Pinus sylvestris* and *Larix decidua* (forest type 2), the return period is slightly smaller as compared to deciduous forests. Furthermore, we also illustrate that differences between forested and non-forested slopes will chiefly depend on forest structure. In this sense, $Nrp_{red}$ is significantly smaller for a clustered tree distribution, gaps or aisles than for a random tree distribution ($p < 0.05$).

The reducing effect of the forest is decreasing with increasing rock volume (Fig. 7; $p < 0.05$). This is especially pronounced for forests with small tree diameters (e.g., forest type 4). Also, $Nrp_{red}$ is significantly reduced in case of zero roughness ($p < 0.05$). A significant difference in $Nrp_{red}$ also exist between soil types 3 and 4 (see Table 1).

According to the final generalized linear model ($GLM_{freq}$), $Nrp_{red}$ is significantly influenced by the cumulative basal area (cbA), rock volume, horizontal forest structure, soil type, soil roughness, and the percentage of conifers present in the forest stand (Table 3). $GLM_{freq}$ has a $R^2$ of 0.80 and a normalized Root Mean Squared Error (nRMSE) of 0.14 with cross-validation for the training data set and 0.16 for the test data set. We also realize that the nRMSE changes only slightly if $GLM_{freq}$ is applied to linear slopes (Table 4).



The variables reported above were also decisive in the regression tree model ($RT_{freq}$). The dataset was first partitioned based on a threshold of 60 m$^2$ ha$^{-1}$ for cbA. In the case where cbA is larger than this value, $Nrp_{red}$ is between 0.3 and 1. At the same time, however, $Nrp_{red}$ clearly decreases in the case that rock volumes become > ~1 m$^3$. On the other hand, and if cbA is smaller than 60.0 m$^2$ ha$^{-1}$, the mean $Nrp_{red}$ drops to 0 (cbA < 28.0 m$^2$ ha$^{-1}$) and 0.5 (cbA > 28.0 m$^2$ ha$^{-1}$ and a rock volume <

0.75 m$^3$). The normalized Root Mean Squared Error (nRMSE) of $RT_{freq}$ is 0.14 with cross-validation for the training dataset and 0.16 for the test dataset. As can be seen from Table 4, the nRMSE is in the same range of values for the linear slopes.

In the case of the field site in Vaujany (Table 5), for which real data exist from experiments, the $GLM_{Freq}$ and the $RT_{Freq}$ models predict $Nrp_{red}$ values of 0.62 and 0.70, respectively, at a distance of 223 m (0.64 measured value) and 0.75 and 0.91,

respectively, at a distance of 324 m (1.0 measured value).

### 3.2.     Effect of forest on rockfall intensity

On the concave slope, the rocks reach energies of up to 3000 kJ at a slope length of 300 and 450 m, respectively. In the last 100 m of the slope (530 m), energies are reduced to zero. Similarly to the rockfall occurrence frequency, rockfall intensity is

distinctly reduced on the forested slopes compared to the non-forested slope (Fig. 5). Again, the reducing effect is decreased with decreasing cbA, increasing rock volume and for the clustered and gappy forest structures (Fig. 6-7). Furthermore, $E95_{red}$ is significantly smaller on slopes with zero roughness (p < 0.05), but does not depend on soil type.

In the final GLM ($GLM_{Int}$), the horizontal forest structure, soil roughness, percentage of conifer trees, cbA, and rock volume have a significant effect on $E95_{red}$. $GLM_{Int}$ has a $R^2$ of 0.84 and a nRMSE of 0.10 with cross-validation for the training data

set and 0.16 for the test data set. If $GLM_{Int}$ is applied to linear slopes, we observe that the nRMSE values increase slightly (Table 4).

In the regression tree model ($RT_{Int}$), cbA was selected as a splitting variable. Figure 7 illustrates that in the case of high cbA (>85.7 m$^2$ha$^{-1}$), $E95_{red}$ is distinctively smaller with a clustered or gappy forest structure (i.e. ~0.75 with a random tree distribution, ~0.55 with aisles, gaps, or clustered tree distribution). The nRMSE of $RT_{Int}$ is 0.10 with cross-validation for the

training data set and 0.09 for the test data set. Similar to $GLM_{Int}$, we observe that the nRMSE of $RT_{Int}$ values hardly changes on linear slopes (Table 4).

### 3.3.     Intensity-frequency curves

Analysis of intensity-frequency distributions of rockfalls depends strongly on the forest cover. In the case of non-forested slopes, the intensity-frequency curve is substantially shifted upward compared to forested slopes at a distance of 300 m downslope from the start area, thereby indicating a higher frequency (intensity) for a given intensity (frequency) (Figure 10).



In other words, the ß and the α coefficients (Eq. 1) of the power law fitted to the intensity-frequency distributions are considerably lower when forest cover is present as compared to non-forested conditions (Table 6).

## 4.       Discussion and conclusion

In this study we investigated the role of forests – in terms of stand density and species composition – on rockfall occurrence at increasing distances from the release area of rockfalls by using a hypothetical slope typical of mountain environments. Based on a large number of simulation runs using different scenarios, we show that rockfall occurrence frequency below forested slopes is reduced between approximately 10 and 90 percent as compared to non-forested slope conditions. Rockfall intensity is also reduced – although to a slightly smaller extent – by 10 and 70 percent. These findings are in agreement with the study of Lopez-Saez et al. (2016) who found a distinct increase in rockfall return periods (e.g., from 143 yrs under non-forested conditions in 1850 to >1000 yrs under recently grown forest in 2013 and for a block volume of 1.2 m$^3$). In this particular case in the Chartreuse massif (France), the disappearance of viticultural landscapes has led to intense (natural) afforestation and can thus be seen as a natural example for the validation of our theoretical results. Similar to our study, Lopez-Saez et al. (2016) also observe that the kinetic energy of rocks clearly decreases at the bottom of the slope and with increasing forested surface, which is again in concert with the findings of our study. Stoffel et al. (2005) investigated spatial and temporal variations of rockfall activity in a protection forest in the Swiss Alps based on dendrogeomorphic data. They reconstructed a decrease in rockfall rates after the recolonization of part of the slope where most of the forest was destroyed after a high magnitude event in 1720. Masuya et al. (2009), on the other hand, did not found a decrease in the number of rockfalls reaching the damage potential at a distance of 350 m from the rockfall source based on three-dimensional simulations taking vegetation probabilistically into account, but an increase in the spread of the rockfalls and lower rock energies. It has to be mentioned that the considered vegetation cover featured relatively small trees and low tree density.

The multivariate statistical models used in this study allowed quantification of the reduction of rockfall frequency and intensity and its prediction under varying forest and slope conditions. Both models (GLM and RT model) revealed that the effect a forest stand has on rockfall will depend clearly on the cumulative basal area (cbA) of trees, the horizontal forest structure, and on the rock volume. We realize that rockfall frequency and intensity are significantly reduced with decreasing cbA and increasing rock volume as well as in clustered or gappy forests, and are now able to quantify these effects. Moreover, the results also demonstrate how the reducing effect of forests is enhanced with increasing soil roughness and soil elasticity. The influence of the two slope parameters was, however, only significant in the GLM, but not in the RT model.

According to the RT models, the forest effect of rockfall frequency appears to depend mainly on cbA and rock volume, whereas cbA and forest structure appear as the most decisive factors for the reduction in rockfall intensity. Rock volume, by contrast, only has a marginal influence on the reducing effect of forest on rockfall intensity (Fig. 8). The maximum reduction of the rockfall energy by forests is reached for volumes between 0.3 and 0.6 m$^3$. This appears to be the optimal combination





between a sufficiently high tree impact probability and rockfall impact energy. For larger rocks, however, impact probability increases further, but the rockfall energy cannot be dissipated during a single tree impact.

The cbA appears to be a good measure of the protection efficacy of forests, as it combines the basal area (which is determined by tree density and tree diameter) with the forested slope length – two parameters which have been promoted as key variables for forest management in previous work (Perret et al., 2004; Berger and Dorren, 2007; Rammer et al., 2015). Based on our results, we recommend a minimum cbA of about 80-90 $m^2\,ha^{-1}$ for rock volumes larger than 1 $m^3$ and a minimum cbA of about 30-40 $m^2\,ha^{-1}$ for volumes smaller than 1 $m^3$. Compared to the minimum threshold of 20 $m^2\,ha^{-1}$ for the basal area of a rockfall protection forest as suggested by Dorren et al. (2015), this corresponds to a forested slope length of 450 m (rock volume >1 $m^3$) and 200 m (rock volume <1 $m^3$), respectively.

According to the RT models, the horizontal forest structure is particularly important when it comes to the reduction of rockfall intensity. We demonstrate that the kinetic energies of rocks are significantly higher in the case of forest stands with a clustered tree structure or in forests with gaps or aisles compared to random tree distribution. The horizontal forest structure, by contrast, is only of secondary importance for the reduction of rockfall frequency and the number of trees which are impacted by the rock in motion will be decisive.

The performance of the implemented statistical prediction models is satisfactory. They yielded relatively low normalized Root Mean Squared Error (nRMSE), also when applying cross-validation. This indicates that the generalization capacity of the models is relatively high and over-fitting unlikely. The application of the models to four different linear slopes with varying slope angles (32°, 35°, 38°, 40°) did not substantially change the nRMSE (Table 4) suggesting that the models are relatively robust with respect to slope angles.

Various factors influence the robustness of the developed models with respect to the applicability to real slopes. The simulated rock volume was limited to 2.0 $m^3$ and therefore they do not necessarily apply to larger volumes. In the GLM, the $Nrp_{red}$ is linearly extrapolated for larger rock volumes, whereas in the RT model a threshold of 2.0 $m^3$ is fixed and the reductive effect of the forest for larger volumes might be overestimated. Furthermore, since we used the rockfall model Rockyfor3D as an important basis for this study, we assume that this model simulates the rockfall process and impacts against trees sufficiently realistic. It has to be considered, however, that the model takes into account two "species" only, being coniferous and broadleaved, for calculating the energy dissipative capacity of trees. In reality, the range of this capacity is much larger and shows huge variations due to, for example, tree vitality, tree anchoring and other site conditions determining tree growth. Additionally, Rockyfor3D uses a simplified stochastic approach to account for different rock shapes. When considering a single rockfall event with a rock shape that does not correspond to standard shapes such as rectangular or spherical, differences between model and reality can be expected.





We could show that the intensity-frequency distributions of rockfall events can be significantly altered below forests compared to non-forested situations. This supports the importance of a coupled consideration of intensity and frequency in order to fully account for the forest effect as it was already reported for other natural hazard processes (Alila et al., 2009). Otherwise, risk analyses are expected to be biased and the risk below forests may be overestimated resulting in over-

5   dimensioned structural protection measures associated with high costs.

Overall, this study substantiates the importance of forests in reducing rockfall risk. The statistical prediction models based on the simulation results for different forest and terrain scenarios allow to quantify this effect and to predict it for other slopes, given the constrains mentioned above. In order to validate these results, the models have to be tested on real slopes. Dendrogeomorphic data on tree impacts (Trappmann and Stoffel, 2013, 2015; Morel et al., 2015) might help evaluation of

10  changes in frequency reduction along the slope depending on the forest structure (Corominas and Moya, 2010). However, serious validation of the difference between forested and non-forested slopes remains difficult since data is missing.

The shown influence of the forest type and structure on rockfall frequency and intensity underlines the importance of forest management aiming at maintenance of its protection function. Disturbances, such as fire, wind, or insects, can temporarily eliminate or at least substantially reduce the protective effect of forests (Maringer et al., 2016; Cordonnier et al., 2008). Also

15  the rockfall process itself, and such as extreme rockfall events, can destroy considerable parts of the forest and, thus, encompass higher rockfall frequency and intensity in the following years (Stoffel et al., 2005).



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





# Figures

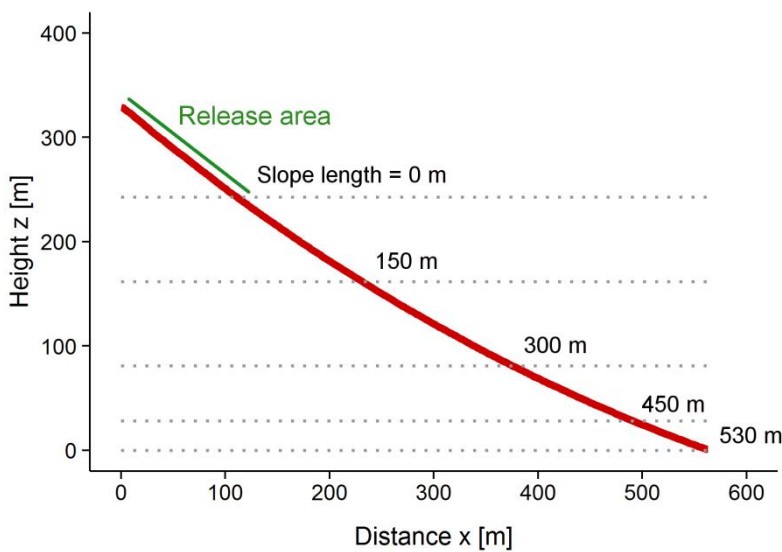

**Fig. 1:** Profile of the virtually constructed digital elevation model (in red) used for the rockfall simulations. Dotted lines indicate the levels at which rockfall occurrence frequency and intensity were evaluated. The rockfall release area is marked in green. The initial fall height of rocks was set to 10 meters above ground.

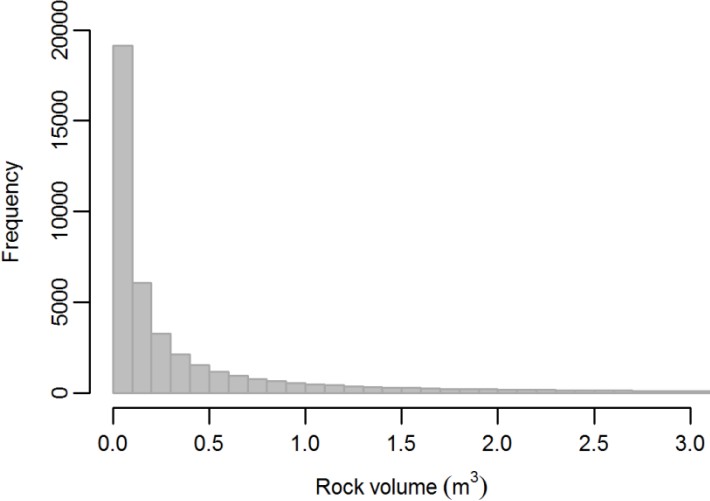

**Fig. 2:** Expected number of rocks released in 1000 yrs on the virtual slope. Calculations are based on a power-law volume-frequency relationship, where ß is the cumulative volume frequency distribution and calculated at 0.463 (Carrea et al., 2014), and where α was set to 12.




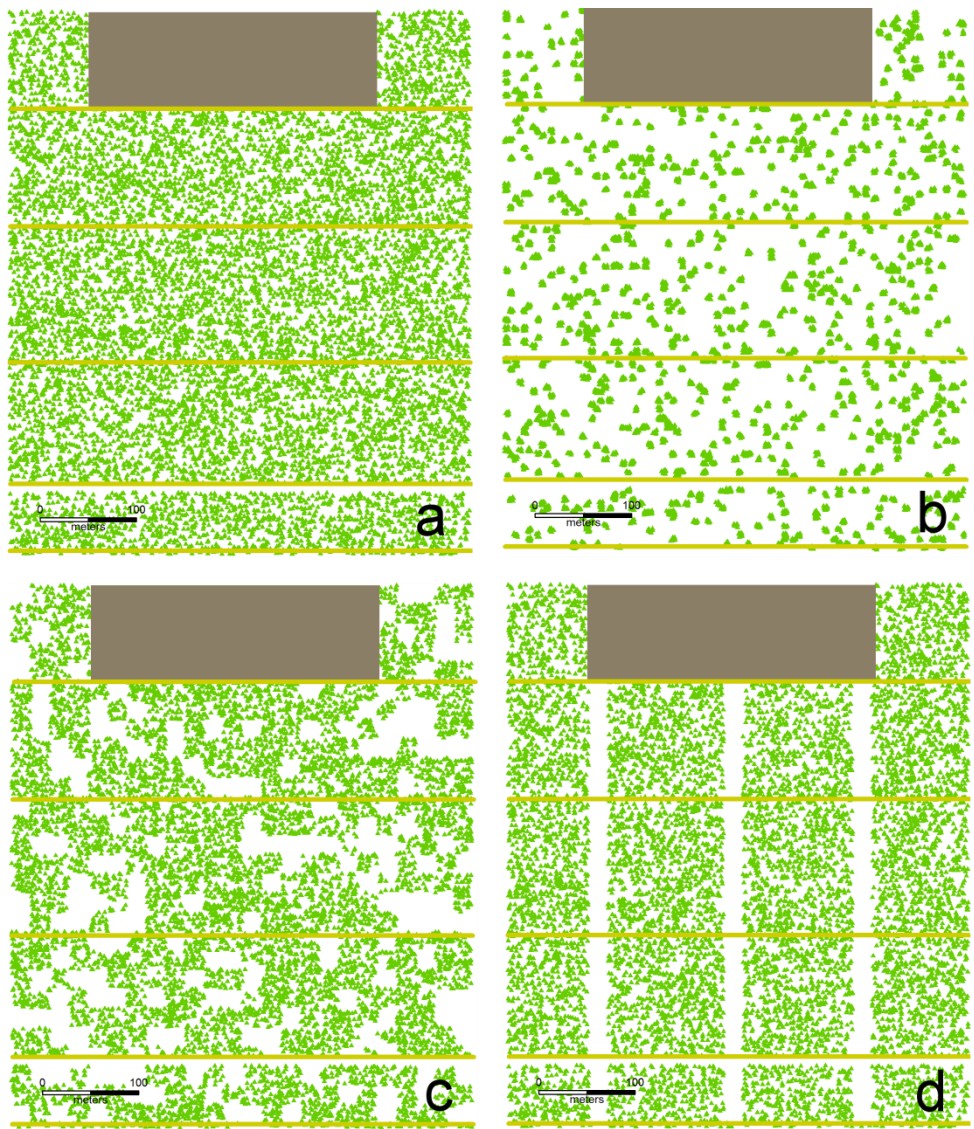

**Fig. 3:** Design of forest structures and release area of rockfalls for simulation runs. For each forest type, we considered four different scenarios regarding the horizontal forest structure. Forest type 1 is illustrated in (a) with a *random* tree distribution and (b) with random distribution of trees in *clusters* of 10 trees; (c) with a distribution of trees with *random gaps* (minimum 20 x 20 m); and in (d) with 3 *aisles* of 20 m in width starting below the release area of rockfall.





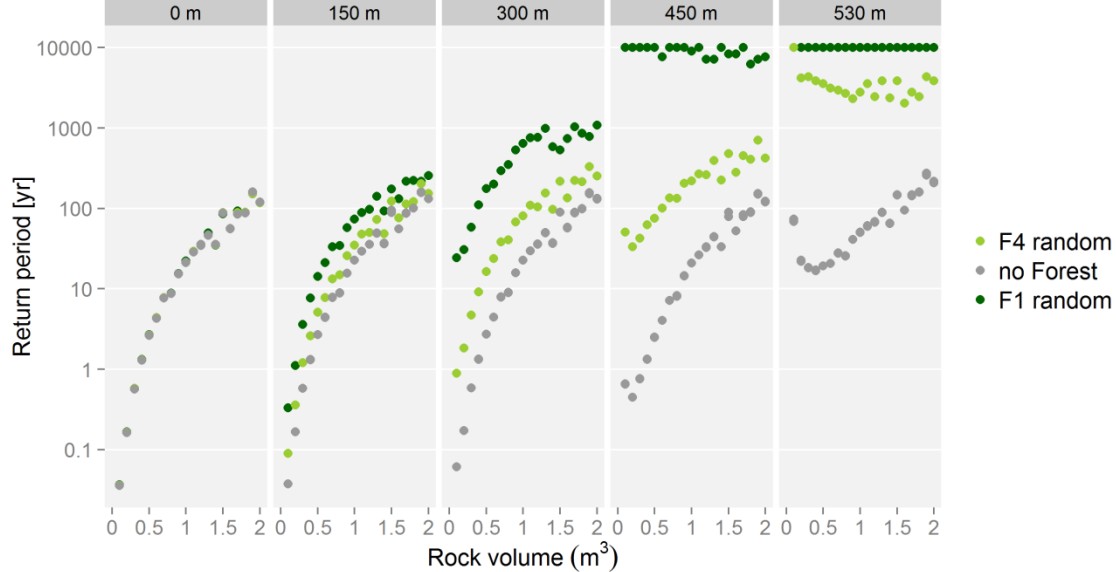

**Figure 4:** Return periods of rockfall (averaged over 100 simulations) at different evaluation zones located at 0-530 m downslope of the release area and for rock volumes ranging from 0.01 to 2.0 m³ under forested (forest type 1: dark green; forest type 4 (F4): light green) and non-forested conditions (grey) with a random tree distribution, soil type 3 and rough slope conditions. Note that the Y-axis is log-transformed.

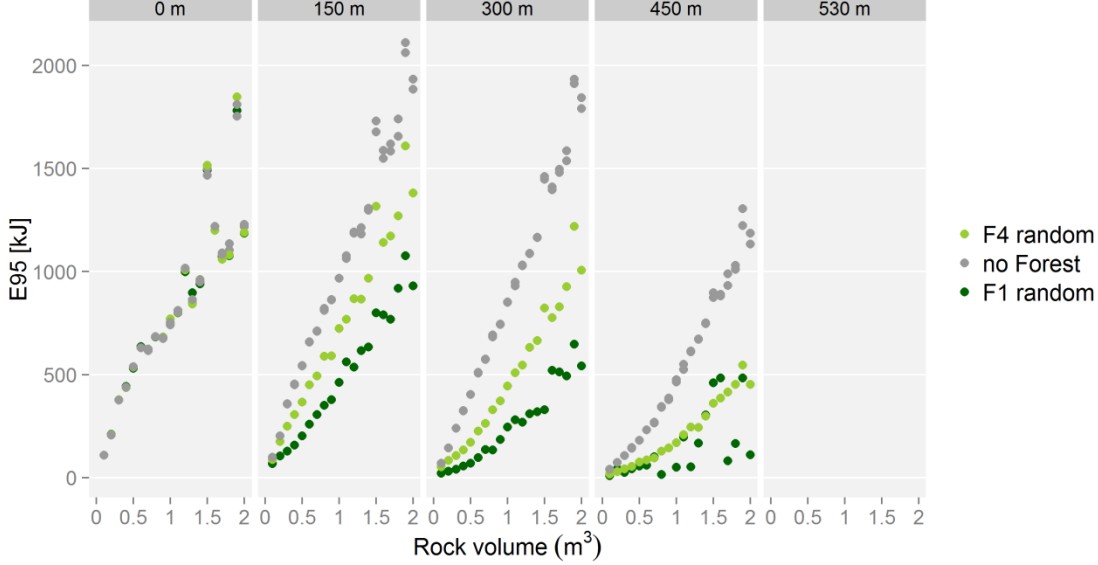

**Figure 5:** Illustration of the 95th percentile of maximum kinetic energies (E95) averaged over 100 simulations and at different evaluation zones located at 0-530 m downslope of the release area. As before, results include a range of rock volumes from 0.01 to 2.0 m³ under




forested (forest type 1 (F1): dark green; forest type 4 (F4): light green) and non-forested conditions (grey) and with a random tree distribution, soil type 3, and rough slope conditions.

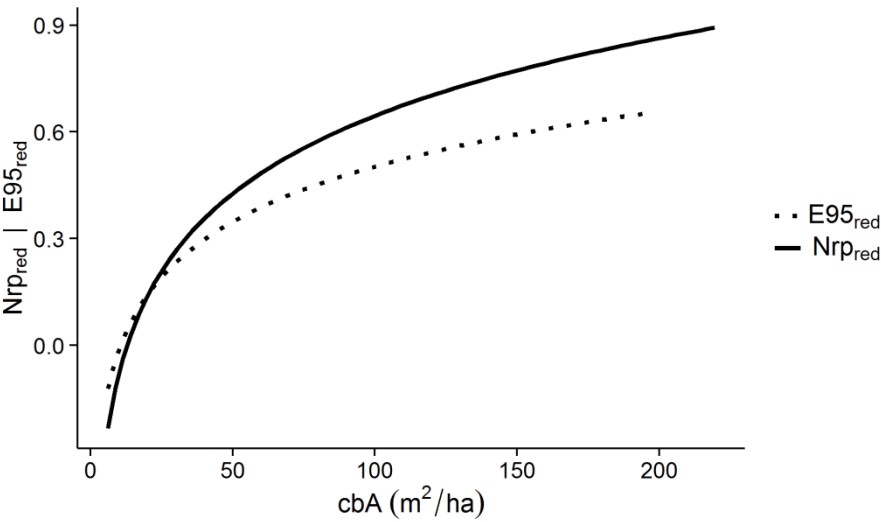

5    **Fig. 6:** $Nrp_{red}$ (solid) and $E95_{red}$ (dotted) based on the simulation of all forest and terrain scenarios on the concave slope and depending on cbA using a logarithmic smoothing function.

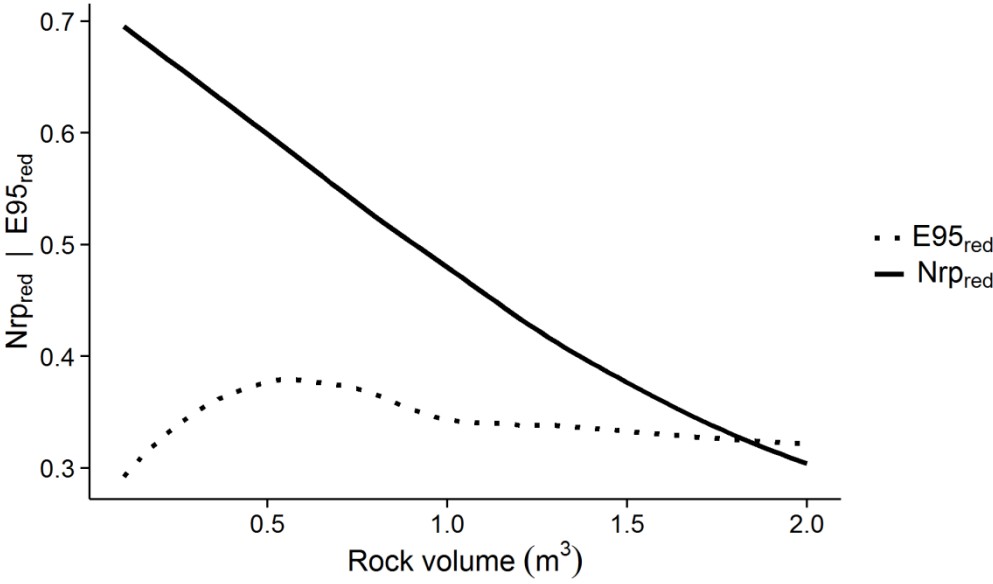





**Fig. 7:** $Nrp_{red}$ (solid) and $E95_{red}$ (dotted) based on the simulation of all forest and terrain scenarios on a concave slope and depending on rock volume using a "loess" smoothing function.




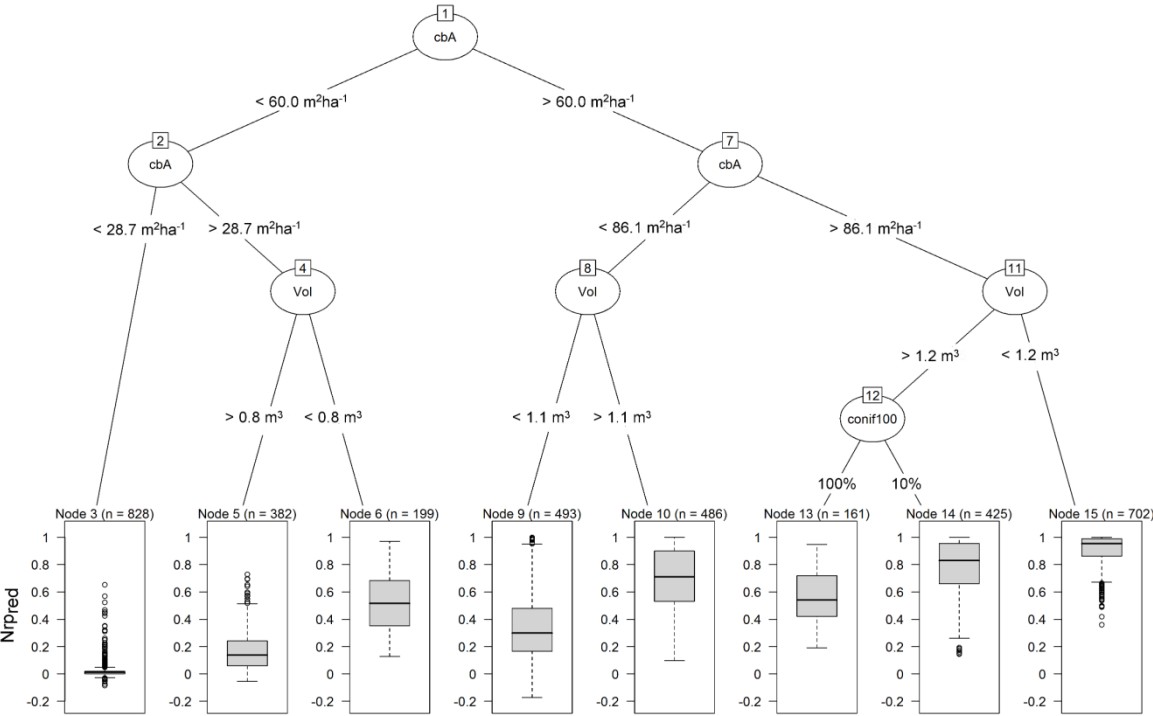

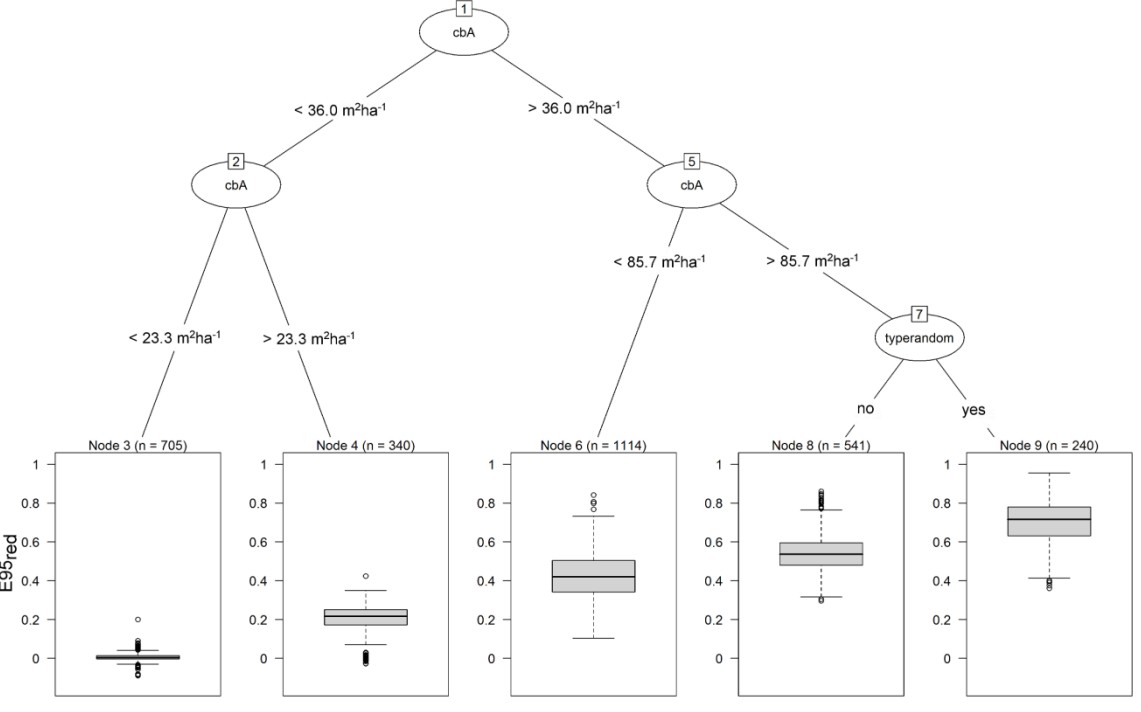





**Figure 8:** Regression tree models were used to predict the reduction in rockfall occurrence frequency ($RT_{Freq}$; above) and the reduction in rockfall intensity ($RT_{Int}$; below) by forests. The models were fitted with a training set representing 75 % of the entire dataset (n=2940) and by applying 3 times 10-fold cross-validation. The nodes represent the splitting variables followed by the applied threshold value. cbA = cumulative basal area [$m^2ha^{-1}$]; Vol = volume [$m^3$]; conif100 = coniferous percent [10, 100 %]; typerandom = random tree distribution [yes, no].

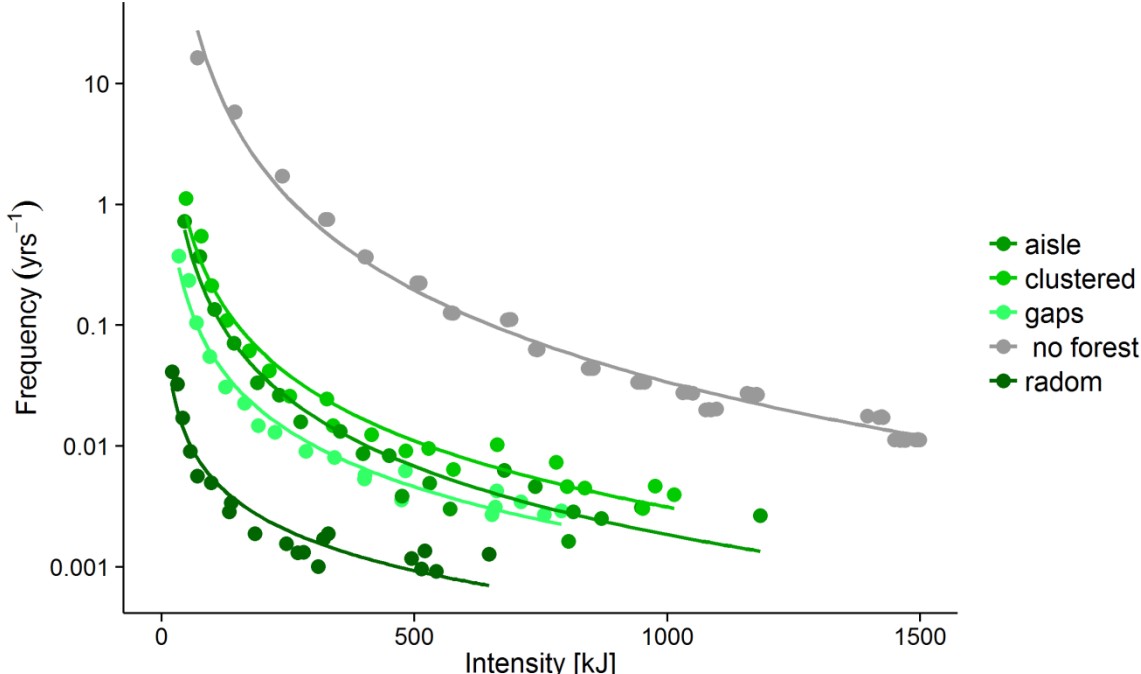

**Figure 9:** Frequency-intensity distributions with fitted power laws at a distance of 300 m from the release area for forest type 1 with different horizontal forest structures and without forest. The intensity is expressed as the 95[th] percentile of the maximum kinetic energy of rocks (E95).



# Tables

**Table 1:** Soil types and roughness used for the different simulation scenarios according to the classification of Dorren (2015). The release area and the forest road were set to no roughness and soil types 5 and 7, respectively, in all scenarios.

|  | **Slope scenarios** |  | **Release area** | **Road** |
|---|---|---|---|---|
| **Soil types** | soil type **3**: scree ø < ~10 cm or medium compact soil with small rock fragments | soil type **4**: talus slope ø > ~10 cm or compact soil with large rock fragments | soil type 5: bedrock with thin weathered material or soil cover | soil type 7: asphalt road |
| **Roughness** | **Rough:** 0.15 (10 %), 0.05 (20 %), 0.01 (70 %) | **No:** 0 m (100 %) | **No:** 0 m (100 %) | **No:** 0 m (100 %) |

**Table 2:** Characteristics of the different forest types used for the rockfall simulations. Values have been taken form the Swiss National Forest Inventory (NFI) datasets published in Messmer (2014).

| Forest type | Definition | Mean number of trees ha$^{-1}$ (with DBH > 12 cm) | Mean DBH [cm] (DBH > 12 cm) | STD DBH | Percentage of conifers [%] |
|---|---|---|---|---|---|
| 1 | Fagus sylvatica 1 | 460 | 33 | 8.36 | 10 |
| 2 | Pinus sylvestris-Larix decidua | 304 | 40 | 10.85 | 100 |
| 3 | Fagus sylvatica 2 | 200 | 33 | 8.36 | 10 |
| 4 | Pole-stand F. Sylvatica | 500 | 21 | 5.00 | 10 |



**Table 3:** Estimated regression coefficients, standard errors, Z-values (i.e. ratio of estimate and standard error), and p-values of the parametric explanatory variables of the general linear model (GLM) for the reduction in rockfall occurrence frequency by forests ($GLM_{Freq}$) and the GLM for the reduction in rockfall intensity ($GLM_{Int}$) by forests. The models were fitted with a training set representing 75 % of the entire dataset (n=2940) applying 3 times a 10-fold cross-validation. Note that $R^2$ $GLM_{Freq}$ = 0.80 and $R^2$ $GLM_{Int}$ = 0.84.

| | Estimate | | Std. Error | | Z-value | | p (>\|z\|) | |
|---|---|---|---|---|---|---|---|---|
| | GLMFreq | GLMInt | GLMFreq | GLMInt | GLMFreq | GLMInt | GLMFreq | GLMInt |
| (Intercept) | -0.58 | -0.57 | 0.014 | 0.009 | -39.23 | -62.30 | <2*10-16 | <2*10-16 |
| Vol | -0.21 | -0.02 | 0.005 | 0.003 | -44.18 | -5.23 | <2*10-16 | 1.8*10-7 |
| log(cbA) | 0.33 | 0.23 | 0.003 | 0.002 | 109.31 | 119.61 | <2*10-16 | <2*10-16 |
| type clustered | -0.1 | -0.03 | 0.007 | 0.005 | -12.03 | -5.99 | <2*10-16 | 2.4*10-9 |
| type gaps | 0.06 | 0.08 | 0.008 | 0.005 | 6.95 | 15.58 | 4.3*10-12 | <2*10-16 |
| type random | 0.06 | 0.09 | 0.008 | 0.005 | 7.66 | 17.72 | 2.8*10-14 | <2*10-16 |
| soil type 4 | -0.05 | | 0.006 | | -7.72 | | 1.5*10-14 | |
| Roughness 2 | -0.09 | -0.02 | 0.006 | 0.004 | -13.63 | -6.18 | <2*10-16 | 7.2*10-10 |
| Conifer percent 100 | -0.10 | -0.09 | 0.008 | 0.005 | -14.26 | -17.54 | <2*10-16 | <2*10-16 |

**Table 4:** Normalized Root Mean Squared Error (nRMSE) of the generalized linear models (GLM) and the regression tree models (RT) predicting $Nrp_{red}$ ($GLM_{Freq}$, $RT_{Freq}$) and $E95_{red}$ ($GLM_{Int}$, $RT_{Int}$) with 3 times 10-fold cross-validation (cv) and for predictions of the test

10 dataset (25 % of the data) and linear slopes with varying slope angle (slope 2-5).

| Model | nRMSE cv | nRMSE test | nRMSE slope 2 (32°) | nRMSE slope 3 (35°) | nRMSE slope 4 (38°) | nRMSE slope 5 (40°) |
|---|---|---|---|---|---|---|



| | | | | | |
|---|---|---|---|---|---|
| **GLM$_{Freq}$** | 14 % | 16 % | 14 % | 15 % | 14 % | 15 % |
| **RT$_{Freq}$** | 14 % | 17 % | 15 % | 16 % | 10 % | 8 % |
| **GLM$_{E95}$** | 10 % | 9 % | 14 % | 13 % | 16 % | 18 % |
| **RT$_{E95}$** | 10 % | 9 % | 10 % | 7 % | 9 % | 11 % |

**Table 5:** Model input parameters and predicted values of Nrp$_{red}$ with the GLM and the RT model as well as the measured value for Nrp$_{red}$ for the study site in Vaujany where (Dorren et al., 2006) performed real-size rockfall experiments.

| Position | Vol [m$^3$] | cbA [m$^2$ ha$^{-1}$] | Forest type | Soil type | Roughness | Nrp$_{red}$ (true) | Nrp$_{red}$ (pred, GLM) | Nrp$_{red}$ (pred, RT) |
|---|---|---|---|---|---|---|---|---|
| Middle slope | 0.5 | 70.5 | Random | 4 | Rough | 0.64 | 0.62 | 0.70 |
| Bottom slope | 0.5 | 102.4 | random | 4 | rough | 1.0 | 0.75 | 0.91 |

**Table 6:** α and ß coefficient and adjusted R$^2$ of the fitted power-laws of the frequency-intensity distributions at a distance of 300 m from the release area for forest type 1 with different horizontal forest structures and without forest.

| Forest structure | α | ß | R$^2$ |
|---|---|---|---|
| No forest | 14.11 | 2.53 | 0.99 |
| Random | -0.17 | 1.09 | 0.93 |
| Clustered | 6.82 | 1.82 | 0.97 |
| Aisle | 6.64 | 1.87 | 0.98 |
| Gaps | 4.29 | 1.56 | 0.96 |

