# Peer review of "Quantifying the effect of forests on frequency and intensity of rockfalls"

_Natural Hazards and Earth System Sciences, 2016_

## Referee Comment (RC1) · Anonymous Referee #1 · 17 Aug 2016

Title

As the paper essentially deals with the impact frequency of blocks and not the occurrence frequency of rockfalls, it is suggested to replace "occurrence frequency" by "impact frequency". Moreover, forest can't influence rockfall occurrence because rockfalls initiate usually in the upper part of slopes, before the falling blocks can be influenced by forest.

General comment

The paper deals with the influence of forest on the impact (or potential impact) frequency and intensity of rock blocks. It is well written and understandable. The influence of forest is clearly demonstrated for a virtual slope, despite a confusion between the rockfall volume distribution and the block volume distribution. Correction of this

confusion needs a revision of the paper. Moreover, the terminology used in the paper is not sufficiently accurate and needs to be modified.

Specific comments

1. The input data which are used for the simulation of blocks propagation are derived from Carrea et al. (2015). But Carrea et al. (2005) give the distribution of the volumes of rockfall events and not of the individual blocks. Similarly, the cited references (Dussauge-Peisser et al., 2002; Malamud et al., 2004) don't deal with the distribution of block volumes. Studies on the distribution of block volumes can be found in the following references: Ruiz-Carulla, R., Corominas, J. & Mavrouli, O. 2015. A methodology to obtain the block size distribution of fragmental rockfall deposits. Landslides, 12: 815–825. Hantz D., Ventroux Q., Rossetti J-P., Berger F. 2016. A new approach of diffuse rockfall hazard. In: Landslides and Engineered Slopes - Aversa et al. (Eds). Associazione Geotecnica Italiana, Rome, Italy, ISBN 978-1-138-02988-0, 1063-1067. This confusion doesn't call into question the results obtained because (a) the simulation has been made on a virtual slope which is not the La Cornalle slope, (b) the values used for the power-law parameters are plausible also for the distribution of block volumes. But the section 2.2 should be rewritten without mentioning the unsuitable references (Dussauge-Peisser et al., 2002; Malamud et al., 2004; Carrea et al., 2005).

2. In the widely used terminology of landslides (Varnes, 1978; Cruden & Varnes, 1996), the word "rock" refers to the material which is implied in the movement and not to the fragments which propagate down ther slope. The fragments implied in a rockfall can be called fragments, particles, projectiles (Bourrier, Dorren, Hungr, 2013), but the word "block" is more commonly used (for example, Ruiz-Carulla, Corominas, Mavrouli, 2016, Comparison of block size distribution in rockfalls). Then I suggest to replace "rock" by "block" in some places. Moreover, a rockfall event consists in two phases: The detachment of a volume of rock from a steep slope and its propagation down the slope (for example, Bourrier, Dorren, Hungr, 2013). When mentioning a frequency, it is important to precise if it is a detachment (or release) frequency or an impact frequency

on an element at risk. The expression "occurrence frequency" used in the manuscript is not explicit, so I suggest to replace it by "release frequency" or "impact frequency".

Page 3, line 6-9 I don't understand what are "reference situations". Could you explain?

Page 4, line 7 Could you please explain why it is necessary to randomly vary the slope angle of each cell?

Page 4, line 9 It should be explained why a vertical fall height of 10 m has been chosen (it is not realistic).

Page 4, line 10 The distances are different from the distances in the Figure 1. For example, the last line must be at a distance of 574-100=474 m from the release area (and not 530 m). This point must be clarified.

Page 5, line 31 The 49 scenarios should be explained: 4 forest types and 4 forest structures give 16 scenarios, but how can one obtain 49 scenarios? The number of slope scenarios doesn't appear clearly in Table 1 (2 soil types and 2 roughness scenarios give 4 slope scenarios!)

Page 7, line 2 Power-laws were fitted for the volume-frequency relation, but the power-law parameters (alpha and beta) are not given in the paper. It would be interesting to compare the beta-values obtained with the beta-value adopted for the initial distribution of block volume. Concerning the intensity-frequency relation, could you please indicate if the E95 values have been averaged over the 100 simulations to obtain the distribution shown in Figure 9? In other words, is the distribution obtained from 239 E95 values (478 m / 2 m) or from 23,900 values (dividing the number of each energy class by 100,000 years)? In my opinion, the most significant distribution in terms of hazard assessment would have been obtained by considering all the energies calculated in all cells rather than only the 95th percentiles (which doesn't contain all the information about the extreme values).

Page 7, line 15-20 The definitions of bA and cbA should be clarified. From the definition

given in the line 17, bA is not an area, but a relative area which reflect the proportion of area which is occupied by trees. It should be called "relative basal area". As bA is dimensionless (m2/ha), it should be multiplied by an area to obtain a total tree area, which influences the impact frequency. I suggest to present the definition as follows: "The latter is defined as the product of the basal area (bA; m2/ha) by the area of the forested slope from the top of the release area to the respective EL, for a width of 100 m." And to define fsL after Equation (7) as the forested slope length. As cbA is an area, it must be expressed in m2, and not in m2ha-1 as written in Equation (7), page 11, line 6-7 and in Table 5. Moreover, in the third member of Equation (7) bA represents the basal area of individual trees and not the (relative) basal area as defined previously. I suggest to remove this third member which is incorrect and unnecessary.

Minor corrections are suggested in the pdf.

Please also note the supplement to this comment:
http://www.nat-hazards-earth-syst-sci-discuss.net/nhess-2016-230/nhess-2016-230-RC1-supplement.pdf

**Supplement:**

[revised manuscript text omitted]

---

## Referee Comment (RC2) · D. Toe (Referee) · 8 Sep 2016

**1   Title**

The actual title of the paper without the term 'occurrence' (see specific comments on terminological approximations) could be used to describe the work done in the paper. However it could be better to highlight the work done on the development of statistical models (meta models) which can predict the protective efficiency of forests against rockfall hazard. The latter result is the innovation in the paper.

**2 General comment**

This paper investigates the influence of forest on block propagation for different scenarios. For each rockfall scenarios, simulations are performed and information on the passing frequency and energy of the blocks is registered at 5 evaluation lines placed along a virtual slope. Using the data from the simulations, different statistical models were developed to predict the effect of forest on rockfall.

I recommend major revisions of the paper for the following reasons:

- Terminological approximations can be found in the paper (Details in specific comments 1).

- Recent references have to be added in the paper, essentially in the introduction, discussion and conclusion sections (Details in specific comments 2).

- Too many objectives are presented in the introduction section (5 are presented P3 L24-32).

- Section 2 (material and methods) has to be restructured to identify clearly the different rockfall scenarios, the input parameters, and data used to build the statistical models presented in the paper. Additional information on the calculation of the indicator of rockfall intensity is requested to evaluate the robustness of the statistical analysis (Details in specific comments 3).

- Presentation of the results has to be improved. Data have to be added in the tables and in the figures. In addition, all the indicators and parameters used in tables have to be presented in the material and methods section (Details in specific comments 4).

- The firs part of the discussion and conclusion has to be improved. Comparison between this study and recent works done to characterize the effect of forest on block propagation would be much appreciated (Details in specific comments 5).

**3   Specific comments**

**3.1   Specific comments 1**

1. The phrase "occurrence frequency" is not adequate. The author can use frequency instead. In addition to this remark, it would be appreciated to use a consistent terminology related with the frequency of blocks passing through an evaluation line: either frequency or return period.

2. P1, L30: "reducing effect of forest" is not adequate. It could be replaced by "protecting effect of forest".

3. P2, L12: "the propagation probability and thus the occurrence frequency" could be modified in reaching probability.

4. P2, L25, L28 "... the annual exceedance frequency ..." this sentence has to be improved.

5. P4, L18: What is a 3D vector? RockyFor3D simulate rockfall propagation using a 3D raster map.

6. P4, L20,L24: "The elasticity of the surface material". Elasticity cannot be used to describe the response of the surface during a block impact. If a soil is elastic no energy dissipation occurred during the impact. It would be preferable to use "the response of the surface material" instead. P10, L28 "Soil elasticity" should be replaced. For example : capacity of the soil to dissipate energy.

[Figure]

7. P5, L30: "aisles" could be replaced by corridors.

**3.2 Specific comments 2**

1. P2, L13: Maybe English references can be added.

2. P2, L16: More recent references can be added on protective measures.

3. P2, L20: More recent references can be added on the protective effect of forests and their maintenance.

4. P3, L15: References can be added on the influence of the forest structure on rockfall propagation.

5. P3, L16: References can be added on tree capacity to absorb energy.

6. P7, L22: References are needed for the calculation of the Spearman correlation coefficients

7. P7, L23: References are needed for the calculation of the AIC

8. P11, L30: Add references

9. P12, L5: Add references

**3.3 Specific comments 3**

The section material and methods has to be improved to present a clear and synthetic description of the rockfall simulation scenarios and the presentation of the input parameters (dedicated sections can be added). In addition, some justifications are missing for the choices of the input values of rockfall simulations.

1. P4: The first paragraph of the material and methods section is not an introduction. It may be placed in a subsection or restructure according to my last comment.

2. P4: The presentation of RockyFor3D can be improved by adding a description of the input raster maps associated to the tree generation. An highlight on the output used for the creation of the statistical models would be appreciated.

3. P4, L33: This sentence should be placed in a section describing all the input parameters used for the simulations.

4. P5, L8: The value chosen for the parameter $\beta$ needs to be added.

5. P5, L10-13: Why didn't you take an interval of block volumes ranging between 0.05 and 5 $m^3$. The justification of the interval: 0.05 and 2 $m^3$, could be improved. In Stockes 2006, forests are presented as having a protection function for block volume until $5 \ m^3$ (Berger et al. 2002; Stoffel et al. 2005).

6. P5, L31: According the element given in the paper, 68 simulation scenarios are identified (2 soiltype * 2 rugosity * 4 forests * 4 horizontal structure + 2 soiltype * 2 rugosity * 1 noForest). The author reach only 49 scenarios.

   Did you define 49 scenarios or 68? It would be appreciated, in both cases, to present them clearly in your paper.

7. P5, 32: Can you give us details about the number of blocks considered for each interval, in particular the interval [1.9-2] $m^3$. Could you create a table to indicate the number of blocks for each interval? This point is particularly important to evaluate the robustness of your statistical analysis.

8. P6 L15-20: It would be preferable to choose either the passing frequency of the block through a line or the return period for all the analysis.
[Figure]

9. P6, L29: A precise description of the method used to calculate the indicator E95red is required to evaluate the robustness of the method. Calculating E95 using the average of E95 values along an evaluation line (these values being averaged over a hundred simulations) cannot be used as a relevant statistical indicator. In addition using the percentile 0.95 of a distribution is only valid for a high number of blocks passing through a line.

How many blocks are used to calculate E95 for the different evaluation lines (especially for large block volumes)?

10. P7, L5-13: In this section a complete description of the two main statistical analyses is required. The statistical model are the main results of this work. Thus, a description of the hypothesis associated to each one is requested.

In addition, for the regression tree models, (P7, L10-11) either you have to explain all the method to select the splitting variable and the impurity reduction or you have to remove those two sentences from the article.

11. P7, L23: The presentation of the GLM method is to short, additional information on the method is requested.

12. P8, L1-3: These two sentences could be placed in the discussion and conclusion section.

**3.4 Specific comments 4**

1. Results of the Wilcoxon rank sum test (methodology section P7 L5) have to be added to the results section.

2. Results of the Spearman correlations coefficient (methodology section P7 L5) have to be added to the results section.

3. P8, L16: In the legend of Fig 4 and 5, the volumes of the blocks ranging between 0.01 to 2 $m^3$ are indicated. A different range is presented in the material and methods section (block volumes ranging between 0.05 to 2 $m^3$).

4. P8, L19-L25: The results presented in Fig 6 and Fig 7 should be improved. The data on which each curve are fitted could be placed in the background of each figure. In addition, the smoothing techniques used to create the curves (Fig 6 and Fig 7) should be detailed in the material and methods section.

5. P8, L31: A simplification of the results presented in table 3 could be done to improve its understanding. In addition, each term used in the table has to be defined in the material and methods section. For example: (Intercept), Vol, GLM-Freq, GLMInt... In table 4 RTFreq and RTE95 are not defined in the material and methods section.

6. P8, L21: Where are the results illustrating the influence of the forest structure? The only results presented are simulation scenarios with random or no forest (Fig4 and Fig5).

7. P9, L1: Why, for the level 8 of the RTFreq model (Fig 8), a volume < 1.1 $m^3$ has a smaller Nrpred value than a volume > 1.1 $m^3$, and why, for the level 11 of the RTFreq model (Fig 8), a volume < 1.2 $m^3$ has a higher Nrpred value than a volume < 1.2 $m^3$? Could you explain the difference?

8. P9, L8-10: This results could be placed in the discussion conclusion section.

9. P9, L30: Which data were used to build the Fig 9?

**3.5 Specific comments 5**

1. P10, L10: The reference given not correct. The result presented in Lopez et al 2016 is : 143 years for forested condition in 1850 to > 2000 years for forested

condition in 2013 for a block volume of 1.2 $m^3$.

2. P10, L15: These two sentences could be moved to the introduction section.

3. P10, L18: The comparison between this work and Matsuya et al 2009 and the explanation of the differences found could be improved.

4. P10, L22-32: This paragraph is currently a description of your results. A comparison between your results and other one from the literature would improve significantly the discussion. Here is a list of recent papers working on similar subjects:

   - Dupire et al. 2016: Novel quantitative indicators to characterize the protective effect of mountain forests against rockfall.
   - Monnet et al. 2016: Suitability of airborne laser scanning for the assessment of forest protection effect against rockfall.
   - Fuhr et al 2015: Protection against rockfall along a maturity gradient in mountain forests
   - Radtke et al. 2014: Managing coppice forests for rockfall protection: lessons from modelling

**3.6 Other comments**

1. P3, L5: "current... reference ", could be replaced by forested and non forested.

2. P3, L17-20: This two sentences should be located in the material and methods section. In the introduction, it would be appreciated to have a short paragraph describing the different rockfall models that can be found in the literature and highlight the few one that take into account the protection effect of the forest.

3. P3, L22-24: Should be placed in the material and methods section.

4. P4, L5: A justification for the choice of a concave profile with slope angle varying between 20 to 40° needs to be added.

5. P4, L6: Why do you add random slope angle variation to your profile? Adding this angle variation comes in conflict with the "controlled conditions" you are looking for P4, L4.

6. P4, L10: Why do you add a road into your slope profile? Adding a road does not appear to be necessary for the analysis done in the paper. Its influence is never presented in the result section nor in the discussion and conclusion section.

7. P5, L29: The reference Fig. 2 have to be replaced by Fig. 3.

8. P7, L16: Why do you choose to calculate cbA using a slope width of 100 m? Did you test other widths and analyse their influence on this indicator?

9. P8: Can you give a ranking of the influence of the different parameters for the 4 statistical models?

10. The author have to pay attention to the consistency of the form of the variable name. Ex : Table 3 GLMFreq and in table 4 $GLM_{Freq}$. See also Table 5.

11. Corrections are suggested in the pdf

Please also note the supplement to this comment:
http://www.nat-hazards-earth-syst-sci-discuss.net/nhess-2016-230/nhess-2016-230-RC2-supplement.pdf

[Figure]

**Supplement:**

[revised manuscript text omitted]

---

## Author Comment (AC1) · 31 Oct 2016

**Reviewer 1**

We thank the reviewer for the very careful revision of our paper and the helpful suggestions. In the following, we respond to the author's comments (blue) and explain changes and adaption we propose for the final article. All comments not specifically addressed in the following list will be adapted in the manuscript as suggested.

**Title**

As the paper essentially deals with the impact frequency of blocks and not the occurrence frequency of rockfalls, it is suggested to replace "occurrence frequency" by "impact frequency". Moreover, forest can't influence rockfall occurrence because rockfalls initiate usually in the upper part of slopes, before the falling blocks can be influenced by forest.

In this study, we define the occurrence frequency as the product of the onset frequency of a block and its propagation probability to a certain position along the slope. The impact frequency is – according to our definition – the product of the occurrence frequency and the presence probability of the element at risk. We agree that the terminology is slightly confusing and, therefore, we try to make it clearer in the introduction. We decided to replace "occurrence frequency" by frequency in the title (see also suggestion of Reviewer 2) in order to avoid confusion.

**Specific comments**

1. The input data which are used for the simulation of blocks propagation are derived from Carrea et al. (2015). But Carrea et al. (2005) give the distribution of the volumes of rockfall events and not of the individual blocks. Similarly, the cited references (Dussauge-Peisser et al., 2002; Malamud et al., 2004) don't deal with the distribution of block volumes. Studies on the distribution of block volumes can be found in the following references: Ruiz-Carulla, R., Corominas, J. & Mavrouli, O. 2015. A methodology to obtain the block size distribution of fragmental rockfall deposits. Landslides, 12: 815–825. Hantz D., Ventroux Q., Rossetti J-P., Berger F. 2016. A new approach of diffuse rockfall hazard. In: Landslides and Engineered Slopes - Aversa et al. (Eds). Associazione Geotecnica Italiana, Rome, Italy, ISBN 978-1-138-02988-0, 1063-1067. This confusion doesn't call into question the results obtained because (a) the simulation has been made on a virtual slope which is not the La Cornalle slope, (b) the values used for the power-law parameters are plausible also for the distribution of block volumes. But the section 2.2 should be rewritten without mentioning the unsuitable references (Dussauge-Peisser et al., 2002; Malamud et al., 2004; Carrea et al., 2005).

Thank you very much for this valuable input. We changed the reference in chapter 2.2 as suggested and excluded the reference for the beta-exponent.

2. In the widely used terminology of landslides (Varnes, 1978; Cruden & Varnes, 1996), the word "rock" refers to the material which is implied in the movement and not to the fragments which propagate down the slope. The fragments implied in a rockfall can be called fragments, particles, projectiles (Bourrier, Dorren, Hungr, 2013), but the word "block" is more commonly used (for example, Ruiz-Carulla, Corominas, Mavrouli, 2016, Comparison of block size distribution in rockfalls). Then I suggest to replace "rock" by "block" in some places. Moreover, a rockfall event consists in two phases: The detachment of a volume of rock from a steep slope and its propagation down the slope (for example, Bourrier, Dorren, Hungr, 2013). When mentioning a frequency, it is important to precise if it is a detachment (or release) frequency or an impact frequency on an element at risk. The expression "occurrence frequency" used in the manuscript

is not explicit, so I suggest to replace it by "release frequency" or "impact frequency".

We agree on that and replaced rock by block as suggested. We further tried to clarify the definition of terms regarding the frequency (see also response 1).

Page 3, line 6-9 I don't understand what are "reference situations". Could you explain?

We refer here to (hypothetical) comparative situations such as other forest scenarios or non-forested situations. We replaced this in the text.

Page 4, line 7 Could you please explain why it is necessary to randomly vary the slope angle of each cell?

We randomly varied the slope angle aiming at more realistic slope conditions. We agree, however, that this is slightly contradictory to the "controlled conditions". Since we re-ran the simulations (see below), we decided to remove the random variation of the slope angle.

Page 4, line 9 It should be explained why a vertical fall height of 10 m has been chosen (it is not realistic).

Since a vertical cliff face is not represented in our virtual slope, we chose an initial fall height of 10 m  which is representative for real rockfall slopes.

Page 4, line 10 The distances are different from the distances in the Figure 1. For example, the last line must be at a distance of 574-100=474 m from the release area (and not 530 m). This point must be clarified.

This is incorrectly written in the text: The distances are measured on the slope (and not horizontally). In Figure 1, we also report the distances along the slope (in the graph).

Page 5, line 31 The 49 scenarios should be explained: 4 forest types and 4 forest structures give 16 scenarios, but how can one obtain 49 scenarios? The number of slope scenarios doesn't appear clearly in table 1.

Correct is a total of 48 scenarios: 4 forest types, 4 forest structures and 3 terrain scenarios (rough + soiltype 3, rough + soiltype 4, zero roughness + soiltype 3 → zero roughness was not combined with soiltype 4).

Page 7, line 2 Power-laws were fitted for the volume-frequency relation, but the powerlaw parameters (alpha and beta) are not given in the paper. It would be interesting to compare the beta-values obtained with the beta-value adopted for the initial distribution of block volume. Concerning the intensity-frequency relation, could you please indicate if the E95 values have been averaged over the 100 simulations to obtain the distribution shown in Figure 9? In other words, is the distribution obtained from 239 E95 values (478 m / 2 m) or from 23,900 values (dividing the number of each energy class by 100,000 years)? In my opinion, the most significant distribution in terms of hazard assessment would have been obtained by considering all the energies calculated in all cells rather than only the 95th percentiles (which doesn't contain all the information about the extreme values).

The alpha and beta coefficients are given in Table 6. We replaced them by the coefficients of the cumulative intensity-frequency distribution allowing for comparison with the initial distribution of block volume (in the old version, it is the non-cumulative distribution). Concerning the intensity values: We averaged the E95 values over the 100 simulations, but we agree that it would be better to calculate the 95-percentile from all energy values yielded

at an evaluation line. Since the second reviewer questions the statistical representativeness of the intensity calculation (due to small block number of large volumes), we decided to re-run the simulations with more blocks (whole release area) and calculated the 95-percentile of the energy values of all blocks per volume passing an evaluation line.

Page 7, line 15-20 The definitions of bA and cbA should be clarified. From the definition given in the line 17, bA is not an area, but a relative area which reflect the proportion of area which is occupied by trees. It should be called "relative basal area". As bA is dimensionless (m2/ha), it should be multiplied by an area to obtain a total tree area, which influences the impact frequency. I suggest to present the definition as follows: "The latter is defined as the product of the basal area (bA; m2/ha) by the area of the forested slope from the top of the release area to the respective EL, for a width of 100 m." And to define fsL after Equation (7) as the forested slope length. As cbA is an area, it must be expressed in m2, and not in m2ha-1 as written in Equation (7), page 11, line 6-7 and in Table 5. Moreover, in the third member of Equation (7) bA represents the basal area of individual trees and not the (relative) basal area as defined previously. I suggest to remove this third member which is incorrect and unnecessary.

It is true that the definition is slightly confusing. cbA is indeed the product of the relative basal area (corrected in the text), which is the sum of the basal areas of individual trees divided by the respective horizontal area (from the bottom of the release area to the respective area), normalized by a slope width of 100 m and multiplied with the forested slope length (measured along the slope).

---

## Author Comment (AC2) · 31 Oct 2016

**Reviewer 2**

We thank the reviewer for the very careful revision of our paper and the helpful suggestions. In the following, we respond to the author's comments (blue) and explain changes and adaption we propose for the final article. All comments not specifically addressed in the following list will be adapted in the manuscript as suggested.

**Title**

The actual title of the paper without the term 'occurrence' (see specific comments on terminological approximations) could be used to describe the work done in the paper. However it could be better to highlight the work done on the development of statistical models (meta models) which can predict the protective efficiency of forests against rockfall hazard. The latter result is the innovation in the paper.

We agree that the term "occurrence" can be deleted in the title. The corresponding terms are introduced and explained later in the text. However, we decided not to name the statistical models in the title, although we agree that the "meta models" are a main innovation of the paper.

**General comments**

Responses to the general comments can be found in the respective specific comments.

**Specific comments 1**

1. The phrase "occurrence frequency" is not adequate. The author can use frequency instead. In addition to this remark, it would be appreciated to use a consistent terminology related with the frequency of blocks passing through an evaluation line: either frequency or return period.

We use the "occurrence frequency" as description of the frequency that a certain point is reached in dependence of the rock release and its propagation. It is thus defined as the product of the "onset frequency" and the "propagation probability". We tried to clarify that in the text.

Regarding the presentation of the results, we agree that a mixing between the presentation of frequencies and return periods may be confusing. For this reason, we only present yearly frequencies in the results and the graphs.

3. P2, L12: "the propagation probability and thus the occurrence frequency" could be modified in reaching probability.

See comment to 3.1.1.

4. P2, L25, L28 "... the annual exceedance frequency ..." this sentence has to be improved.

We tried to formulate the sentence and the whole paragraph more clearly.

5. P4, L18: What is a 3D vector? RockyFor3D simulate rockfall propagation using a 3D raster map.

This is a slightly unclear formulation. Therefore, we rewrote the sentence.

6. P4, L20,L24: "The elasticity of the surface material". Elasticity cannot be used to describe the response of the surface during a block impact. If a soil is elastic no energy dissipation occurred during the impact. It would be preferable to use "the response of the surface material" instead. P10, L28 "Soil elasticity" should be replaced. For example : capacity of the soil to dissipate energy.

Thank you for this suggestion which we implemented in the text.

7. P5, L30: "aisles" could be replaced by corridors.

After doing additional research, we replaced the term "aisles" by "clearcut strips", which is apparently the most adapted forestry term.

**Specific comments 2**

References were added where deemed necessary.

**Specific comments 3**

1. P4: The first paragraph of the material and methods section is not an introduction. It may be placed in a subsection or restructure according to my last comment.

We placed the first paragraph of section 2 in a separate subsection.

2. P4: The presentation of RockyFor3D can be improved by adding a description of the input raster maps associated to the tree generation. An highlight on the output used for the creation of the statistical models would be appreciated.

A short description of the tree input data is already present. The output used for the statistical models is described in section 2.5.

4. P5, L8: The value chosen for the parameter needs to be added.

We added the value for β.

5. P5, L10-13: Why didn't you take an interval of block volumes ranging between 0.05 and 5 $m3$. The justification of the interval: 0.05 and 2 $m3$, could be improved. In Stockes 2006, forests are presented as having a protection function for block volume until $5\ m3$ (Berger et al. 2002; Stoffel et al. 2005).

We decided to simulate only block volumes between 0.05 and 2 m$^3$ because certain references mention a volume of 2 m$^3$ as limiting volume for the protective function of forests (reference changed in the article in order to avoid confusion). Blocks with volumes between 0.05 and 2 m$^3$ are most risk relevant as they exhibit high occurrence frequencies.

6. P5, L31: According the element given in the paper, 68 simulation scenarios are identified (2 soiltype * 2 rugosity * 4 forests * 4 horizontal structure + 2 soiltype *

2 rugosity * 1 noForest). The author reach only 49 scenarios.
Did you define 49 scenarios or 68? It would be appreciated, in both cases, to present them clearly in your paper.

Correct is a total of 48 scenarios: 4 forest types, 4 forest structures and 3 terrain scenarios (rough + soiltype 3, rough + soiltype 4, zero roughness + soiltype 3 > zero roughness was not combined with soiltype 4). We tried to better explain them in the text.

7. P5, 32: Can you give us details about the number of blocks considered for each interval, in particular the interval [1.9-2] $m3$. Could you create a table to indicate the number of blocks for each interval? This point is particularly important to evaluate the robustness of your statistical analysis.

Due to your justified questioning of the statistical robustness of the simulation results for larger volumes, we decided to run new simulations with a higher and more robust number of blocks. We simulated the whole release area (7500 cells) with 50 simulations per block for all block volumes. Based on these simulations, we calculated the propagation probabilities of the blocks (described in section 2.2.) and multiplied them with the onset frequency in order to obtain the occurrence frequencies. We further evaluated the 95-percentile of the maximum energies of all blocks passing an evaluation line (not as the mean of the 95-percentile of all cells).

8. P6 L15-20: It would be preferable to choose either the passing frequency of the block through a line or the return period for all the analysis.

See response "Specific comments 1, 1."

9. P6, L29: A precise description of the method used to calculate the indicator E95red is required to evaluate the robustness of the method. Calculating E95 using the average of E95 values along an evaluation line (these values being averaged over a hundred simulations) cannot be used as a relevant statistical indicator. In addition using the percentile 0.95 of a distribution is only valid for a high number of blocks passing through a line.
How many blocks are used to calculate E95 for the different evaluation lines (especially for large block volumes)?

See response "Specific comments 3, 7."

10. P7, L5-13: In this section a complete description of the two main statistical analyses is required. The statistical model are the main results of this work. Thus, a description of the hypothesis associated to each one is requested.
In addition, for the regression tree models, (P7, L10-11) either you have to explain all the method to select the splitting variable and the impurity reduction or you have to remove those two sentences from the article.

11. P7, L23: The presentation of the GLM method is to short, additional information on the method is requested.

We agree that the statistical models are the main results of the study and a special focus should be placed on them. However, since the two applied multivariate models are well-established and well-described in literature, we describe them briefly, however with sufficient literature references for more detail.

**Specific comments 4**

1. Results of the Wilcoxon rank sum test (methodology section P7 L5) have to be added to the results section.

P-values of the Wilcoxon rank sum test are reported in brackets for the respective variables. In order to keep the result section short and concise, we do not report exact p-values and parameters of the Wilcoxon rank sum tests.

2. Results of the Spearman correlations coefficient (methodology section P7 L5) have to be added to the results section.

The Spearman correlation coefficients were calculated in order to exclude strongly correlated explanatory variables in the multivariate statistical models. This was not clearly described in the article and thus adapted. In order to keep the result section focused, we do not report them in detail.

3. P8, L16: In the legend of Fig 4 and 5, the volumes of the blocks ranging between 0.01 to 2 $m3$ are indicated. A different range is presented in the material and methods section (block volumes ranging between 0.05 to 2 $m3$).

Thank you for this indication. Volumes ranging from 0.05 to 2.0 $m^3$ were simulated.

4. P8, L19-L25: The results presented in Fig 6 and Fig 7 should be improved. The data on which each curve are fitted could be placed in the background of each figure. In addition, the smoothing techniques used to create the curves (Fig 6 and Fig 7) should be detailed in the material and methods section.

We decided to plot only the fitted curves in order to highlight and summarize the main tendency of the data in a striking way. However, we understand your point and we will re-examine several alternatives for plotting the underlying data.

5. P8, L31: A simplification of the results presented in table 3 could be done to improve its understanding. In addition, each term used in the table has to be defined in the material and methods section. For example: (Intercept), Vol, GLMFreq, GLMInt... In table 4 RTFreq and RTE95 are not defined in the material and methods section.

You are right that certain abbreviations are not introduced in the material and methods and the table caption, respectively. We corrected this. The presented parameters of the GLM (e.g. Z-value), however, correspond to the standard representation of the results of a GLM and thus, we omit to explain them in more detail.

6. P8, L21: Where are the results illustrating the influence of the forest structure?
The only results presented are simulation scenarios with random or no forest
(Fig4 and Fig5).

We decided to show results of two different forest types and no forest in fig. 4 and 5 as we do
not want to overload the figure. The influence of the forest structure is reported in the text
and can be seen in fig. 9 (intensity-frequency curves) as well as the statistical models.

7. P9, L1: Why, for the level 8 of the RTFreq model (Fig 8), a volume < 1.1 $m3$
has a smaller Nrpred value than a volume > 1.1 $m3$, and why, for the level 11 of
the RTFreq model (Fig 8), a volume < 1.2 $m3$ has a higher Nrpred value than a
volume < 1.2 $m3$? Could you explain the difference?

There was a mistake in the initial graph at level 8: There is a smaller $Nrp_{red}$ value for volume
> 1.1 m$^3$ than for volumes < 1.1 m$^3$. Level 11, however, is correct: $Nrp_{red}$ is smaller for
volumes > 1.2 m$^3$ than for volumes < 1.2 m$^3$. In case of volumes > 1.2 m3, $Nrp_{red}$ is smaller
for confiers (conif 100%) than deciduous forests (conif 10%).

9. P9, L30: Which data were used to build the Fig 9?

Description of figure will be corrected according to "Specific comments 3, 5."

**Specific comments 5**

4. P10, L22-32: This paragraph is currently a description of your results. A comparison
between your results and other one from the literature would improve
significantly the discussion. Here is a list of recent papers working on similar
subjects:

We thank for the interesting literature suggestions and integrated part of them in our
discussion. However, we did not substantially change the mentioned paragraph as we do not
only describe the results but also highlight and summarize their significance. A discussion
and comparison with other literature follow in the next paragraph(s).

**Other comments**

2. P3, L17-20: This two sentences should be located in the material and methods
section. In the introduction, it would be appreciated to have a short paragraph
describing the different rockfall models that can be found in the literature and
highlight the few one that take into account the protection effect of the forest.

We made reference to Volkwein et al. 2011 who summarize and describe in detail a wide
variety of existing rockfall models.

4. P4, L5: A justification for the choice of a concave profile with slope angle varying
between 20 to 40° needs to be added.

We chose a concave profile as this corresponds to typical and probably most frequent slope
geometries of rockfall slopes. In order to test the influence of the slope profile on the results,

we validated the statistical models with results of simulations on linear slopes with slope angles of 32°, 35°, 38° and 40°.

5. P4, L6: Why do you add random slope angle variation to your profile? Adding this angle variation comes in conflict with the "controlled conditions" you are looking for P4, L4.

6. P4, L10: Why do you add a road into your slope profile? Adding a road does not appear to be necessary for the analysis done in the paper. Its influence is never presented in the result section nor in the discussion and conclusion section.

We randomly varied the slope angle of each cell aiming at more realistic conditions. We agree, however, that this is slightly contradictory to the "controlled conditions". For this reason, we decided to run the new simulations without slope angle variation and we also omitted the road in the new slope as it is indeed not necessary for the presented analysis.

8. P7, L16: Why do you choose to calculate cbA using a slope width of 100 m? Did you test other widths and analyse their influence on this indicator?

The slope width of 100 m is used to normalize the cumulative basal area for a given runout distance. As such, it does not make any difference whether we use 1, 10 or 100 m.

9. P8: Can you give a ranking of the influence of the different parameters for the 4 statistical models?

The p-values of the parameters and the position in the regression tree can be regarded as ranking. This, however, has to be interpreted carefully, since the multivariate models reflect interactions of the parameters and a ranking does not necessarily make sense.

---

## Author Response (AR1)

**Reviewer 1**

We thank the reviewer for the very careful revision of our paper and the helpful suggestions. In the following, we respond to the author's comments (blue) and explain changes and adaption we made in the final manuscript. All comments not specifically addressed in the following list are adapted in the manuscript as suggested.

**Title**

RC1.1) As the paper essentially deals with the impact frequency of blocks and not the occurrence frequency of rockfalls, it is suggested to replace "occurrence frequency" by "impact frequency". Moreover, forest can't influence rockfall occurrence because rockfalls initiate usually in the upper part of slopes, before the falling blocks can be influenced by forest.

In this study, we define the occurrence frequency as the product of the onset frequency of a block and its propagation probability to a certain position along the slope. The impact frequency is – according to our definition – the product of the occurrence frequency and the presence probability of the element at risk. We agree that the terminology is slightly confusing and, therefore, we try to make it clearer in the introduction (P2 L13, L26ff). We decided to replace "occurrence frequency" by frequency in the title (see also suggestion of Reviewer 2) in order to avoid confusion.

**Specific comments**

RC1.2) The input data which are used for the simulation of blocks propagation are derived from Carrea et al. (2015). But Carrea et al. (2005) give the distribution of the volumes of rockfall events and not of the individual blocks. Similarly, the cited references (Dussauge-Peisser et al., 2002; Malamud et al., 2004) don't deal with the distribution of block volumes. Studies on the distribution of block volumes can be found in the following references: Ruiz-Carulla, R., Corominas, J. & Mavrouli, O. 2015. A methodology to obtain the block size distribution of fragmental rockfall deposits. Landslides, 12: 815–825. Hantz D., Ventroux Q., Rossetti J-P., Berger F. 2016. A new approach of diffuse rockfall hazard. In: Landslides and Engineered Slopes - Aversa et al. (Eds). Associazione Geotecnica Italiana, Rome, Italy, ISBN 978-1-138-02988-0, 1063-1067. This confusion doesn't call into question the results obtained because (a) the simulation has been made on a virtual slope which is not the La Cornalle slope, (b) the values used for the power-law parameters are plausible also for the distribution of block volumes. But the section 2.2 should be rewritten without mentioning the unsuitable references (Dussauge-Peisser et al., 2002; Malamud et al., 2004; Carrea et al., 2005).

Thank you very much for this valuable input. We changed the reference in chapter 2.2 as suggested and excluded the reference for the beta-exponent.

RC1.3) In the widely used terminology of landslides (Varnes, 1978; Cruden & Varnes, 1996), the word "rock" refers to the material which is implied in the movement and not to the fragments which propagate down the slope. The fragments implied in a rockfall can be called fragments, particles, projeciles (Bourrier, Dorren, Hungr, 2013), but the word "block" is more commonly used (for example, Ruiz-Carulla, Corominas, Mavrouli, 2016, Comparison of block size distribution in rockfalls). Then I suggest to replace "rock" by "block" in some places. Moreover, a rockfall event consists in two phases: The detachment of a volume of rock from a steep slope and its propagation down the slope (for example, Bourrier, Dorren, Hungr, 2013). When mentioning a frequency, it is important to precise if it is a detachment (or release) frequency or an impact frequency on an element at risk. The expression "occurrence frequency" used in the manuscript

is not explicit, so I suggest to replace it by "release frequency" or "impact frequency".

We agree on that and replaced rock by block as suggested. We further tried to clarify the definition of terms regarding the frequency (P2 L26ff; P6 L9ff).

RC1.4) Page 3, line 6-9 I don't understand what are "reference situations". Could you explain?

We refer here to (hypothetical) comparative situations such as other forest scenarios or non-forested situations. We replaced this in the text (P3 L9).

RC1.5) Page 4, line 7 Could you please explain why it is necessary to randomly vary the slope angle of each cell?

We randomly varied the slope angle aiming at more realistic slope conditions. We agree, however, that this is slightly contradictory to the "controlled conditions". Since we re-ran the simulations (see below), we decided to remove the random variation of the slope angle.

RC1.6) Page 4, line 9 It should be explained why a vertical fall height of 10 m has been chosen (it is not realistic).

Since a vertical cliff face is not represented in our virtual slope, we chose an initial fall height of 10 m which is representative for real rockfall slopes.

RC1.7) Page 4, line 10 The distances are different from the distances in the Figure 1. For example, the last line must be at a distance of 574-100=474 m from the release area (and not 530 m). This point must be clarified.

This is incorrectly written in the text: The distances are measured on the slope (and not horizontally). In Figure 1, we also report the distances along the slope (in the graph).

RC1.8) Page 5, line 31 The 49 scenarios should be explained: 4 forest types and 4 forest structures give 16 scenarios, but how can one obtain 49 scenarios? The number of slope scenarios doesn't appear clearly in table 1.

Correct is a total of 48 scenarios: 4 forest types, 4 forest structures and 3 terrain scenarios (rough + soiltype 3, rough + soiltype 4, zero roughness + soiltype 3 → zero roughness was not combined with soiltype 4).

RC1.9) Page 7, line 2 Power-laws were fitted for the volume-frequency relation, but the powerlaw parameters (alpha and beta) are not given in the paper. It would be interesting to compare the beta-values obtained with the beta-value adopted for the initial distribution of block volume. Concerning the intensity-frequency relation, could you please indicate if the E95 values have been averaged over the 100 simulations to obtain the distribution shown in Figure 9? In other words, is the distribution obtained from 239 E95 values (478 m / 2 m) or from 23,900 values (dividing the number of each energy class by 100,000 years)? In my opinion, the most significant distribution in terms of hazard assessment would have been obtained by considering all the energies calculated in all cells rather than only the 95th percentiles (which doesn't contain all the information about the extreme values).

We only fitted power-laws to the intensity-frequency distribution at a slope length of 300 m and report the parameters in Table 6. Since the second reviewer questions the statistical representativeness of the intensity calculation (due to small block number of large volumes), we decided to re-run the simulations for the whole release area (50 blocks per cell) and

calculated the occurrence frequency in a second step by multiplying the onset frequency with the propagation probability (see P6 L9ff for detailed description). The E90 values in the distribution shown in Figure 9 were obtained based on all energy values of the blocks passed through the respective line.

RC1.10) Page 7, line 15-20 The definitions of bA and cbA should be clarified. From the definition given in the line 17, bA is not an area, but a relative area which reflect the proportion of area which is occupied by trees. It should be called "relative basal area". As bA is dimensionless (m2/ha), it should be multiplied by an area to obtain a total tree area, which influences the impact frequency. I suggest to present the definition as follows: "The latter is defined as the product of the basal area (bA; m2/ha) by the area of the forested slope from the top of the release area to the respective EL, for a width of 100 m." And to define fsL after Equation (7) as the forested slope length. As cbA is an area, it must be expressed in m2, and not in m2ha-1 as written in Equation (7), page 11, line 6-7 and in Table 5. Moreover, in the third member of Equation (7) bA represents the basal area of individual trees and not the (relative) basal area as defined previously. I suggest to remove this third member which is incorrect and unnecessary.

It is true that the definition is slightly confusing. cbA is indeed the product of the relative basal area (corrected in the text, P7 L12ff), which is the sum of the basal areas of individual trees divided by the respective horizontal area (from the bottom of the release area to the respective area), normalized by a slope width of 100 m and multiplied with the forested slope length (measured along the slope).

**Reviewer 2**

We thank the reviewer for the very careful revision of our paper and the helpful suggestions. In the following, we respond to the author's comments (blue) and explain changes and adaption we made in the final manuscript. All comments not specifically addressed in the following list are adapted in the manuscript as suggested.

**Title**

RC2.1) The actual title of the paper without the term 'occurrence' (see specific comments on terminological approximations) could be used to describe the work done in the paper. However it could be better to highlight the work done on the development of statistical models (meta models) which can predict the protective efficiency of forests against rockfall hazard. The latter result is the innovation in the paper.

We agree that the term "occurrence" can be deleted in the title. The corresponding terms are introduced and explained later in the text. However, we decided not to name the statistical models in the title, although we agree that the "meta models" are a main innovation of the paper.

**General comments**

Responses to the general comments can be found in the respective specific comments.

**Specific comments 1**

RC2.2) The phrase "occurrence frequency" is not adequate. The author can use frequency instead. In addition to this remark, it would be appreciated to use a consistent terminology related with the frequency of blocks passing through an evaluation line: either frequency or return period.

We use the "occurrence frequency" as description of the frequency that a certain point is reached in dependence of the rock release and its propagation. It is thus defined as the product of the "onset frequency" and the "propagation probability". We tried to clarify that in the text (P2 L14; P6 L9ff).

Regarding the presentation of the results, we agree that a mixing between the presentation of frequencies and return periods may be confusing. For this reason, we only present yearly frequencies in the results and the graphs (figure 4).

RC2.3) P2, L12: "the propagation probability and thus the occurrence frequency" could be modified in reaching probability.

See comment to RC2.2.

RC2.4) P2, L25, L28 "... the annual exceedance frequency ..." this sentence has to be improved.

We tried to formulate the sentence and the whole paragraph more clearly (P2 L26ff).

RC2.5) P4, L18: What is a 3D vector? RockyFor3D simulate rockfall propagation using a 3D raster map.

This is a slightly unclear formulation. Therefore, we rewrote the sentence (P4 L20).

RC2.6) P4, L20,L24: "The elasticity of the surface material". Elasticity cannot be used to describe the response of the surface during a block impact. If a soil is elastic no energy dissipation occurred during the impact. It would be preferable to use "the response of the surface material" instead. P10, L28 "Soil elasticity" should be replaced. For example : capacity of the soil to dissipate energy.

Thank you for this suggestion which we implemented in the text.

RC2.7) P5, L30: "aisles" could be replaced by corridors.

Corridors would also be a suitable term. However, we decided to keep "aisle":

**Specific comments 2**

RC2.8)

References were added where deemed necessary.

**Specific comments 3**

RC2.9) P4: The first paragraph of the material and methods section is not an introduction. It may be placed in a subsection or restructure according to my last comment.

We placed the first paragraph of section 2 in a separate subsection.

RC2.10) P4: The presentation of RockyFor3D can be improved by adding a description of the input raster maps associated to the tree generation. An highlight on the output used for the creation of the statistical models would be appreciated.

A short description of the tree input data is already present (section 2.2). The output used for the statistical models is described in section 2.5.

RC2.11) P5, L8: The value chosen for the parameter needs to be added.

We added the value for β.

RC2.12) P5, L10-13: Why didn't you take an interval of block volumes ranging between 0.05 and 5 m3. The justification of the interval: 0.05 and 2 m3, could be improved. In Stockes 2006, forests are presented as having a protection function for block volume until 5  m3 (Berger et al. 2002; Stoffel et al. 2005).

We decided to simulate only block volumes between 0.05 and 2 $m^3$ because certain references mention a volume of 2 $m^3$ as limiting volume for the protective function of forests (reference changed in the article in order to avoid confusion; P5 L11ff). Blocks with volumes between 0.05 and 2 $m^3$ are most risk relevant as they exhibit high occurrence frequencies.

RC2.13) P5, L31: According the element given in the paper, 68 simulation scenarios are

identified (2 soiltype * 2 rugosity * 4 forests * 4 horizontal structure + 2 soiltype * 2 rugosity * 1 noForest). The author reach only 49 scenarios.
Did you define 49 scenarios or 68? It would be appreciated, in both cases, to present them clearly in your paper.

Correct is a total of 48 scenarios: 4 forest types, 4 forest structures and 3 terrain scenarios (rough + soiltype 3, rough + soiltype 4, zero roughness + soiltype 3 > zero roughness was not combined with soiltype 4). We tried to better explain them in the text (section 2.4) and Table 1.

RC2.14) P5, 32: Can you give us details about the number of blocks considered for each interval, in particular the interval [1.9-2] m3. Could you create a table to indicate the number of blocks for each interval? This point is particularly important to evaluate the robustness of your statistical analysis.

Due to your justified questioning of the statistical robustness of the simulation results for larger volumes, we decided to run new simulations with a higher and more robust number of blocks. We simulated the whole release area (7500 cells) with 50 simulations per block for all block volumes. Based on these simulations, we calculated the propagation probabilities of the blocks (described in section 2.2.) and multiplied them with the onset frequency in order to obtain the occurrence frequencies. We further evaluated the 90-percentile of the maximum energies of all blocks passing an evaluation line (not as the mean of the 90-percentile of all cells).

RC2.15) P6 L15-20: It would be preferable to choose either the passing frequency of the block through a line or the return period for all the analysis.

See response "Specific comments 1, 1."

RC2.16) P6, L29: A precise description of the method used to calculate the indicator E95red is required to evaluate the robustness of the method. Calculating E95 using the average of E95 values along an evaluation line (these values being averaged over a hundred simulations) cannot be used as a relevant statistical indicator. In addition using the percentile 0.95 of a distribution is only valid for a high number of blocks passing through a line.
How many blocks are used to calculate E95 for the different evaluation lines (especially for large block volumes)?

See response "Specific comments 3, 7."

RC2.17) P7, L5-13: In this section a complete description of the two main statistical analyses is required. The statistical model are the main results of this work. Thus, a description of the hypothesis associated to each one is requested.
In addition, for the regression tree models, (P7, L10-11) either you have to explain all the method to select the splitting variable and the impurity reduction or you have to remove those two sentences from the article.

RC2.18) P7, L23: The presentation of the GLM method is to short, additional information on the method is requested.

We agree that the statistical models are the main results of the study and a special focus should be placed on them. However, since the two applied multivariate models are well-established and well-described in literature, we describe them briefly, providing relevant literature references for more detail.

**Specific comments 4**

RC2.19) Results of the Wilcoxon rank sum test (methodology section P7 L5) have to be added to the results section.

P-values of the Wilcoxon rank sum test are reported in brackets for the respective variables. In order to keep the result section short and concise, we do not report exact p-values and parameters of the Wilcoxon rank sum tests.

RC2.20) Results of the Spearman correlations coefficient (methodology section P7 L5) have to be added to the results section.

The Spearman correlation coefficients were calculated in order to exclude strongly correlated explanatory variables in the multivariate statistical models. This was not clearly described in the article and thus adapted.  In order to keep the result section focused, we do not report them in detail.

RC2.21) P8, L16: In the legend of Fig 4 and 5, the volumes of the blocks ranging between 0.01 to 2 m3 are indicated. A different range is presented in the material and methods section (block volumes ranging between 0.05 to 2 m3).

Thank you for this indication. Volumes ranging from 0.05 to 2.0 m$^3$ were simulated.

RC2.22) P8, L19-L25: The results presented in Fig 6 and Fig 7 should be improved. The data on which each curve are fitted could be placed in the background of each figure. In addition, the smoothing techniques used to create the curves (Fig 6 and Fig 7) should be detailed in the material and methods section.

We decided to plot only the fitted curves in order to highlight and summarize the main tendency of the data in a striking way. However, we understand your point and, thus, we added the 10%- and 90% confidence intervals.

RC2.23) P8, L31: A simplification of the results presented in table 3 could be done to improve its understanding. In addition, each term used in the table has to be defined in the material and methods section. For example: (Intercept), Vol, GLMFreq, GLMInt... In table 4 RTFreq and RTE95 are not defined in the material and methods section.

You are right that certain abbreviations are not introduced in the material and methods and the table caption, respectively. We corrected this. The presented parameters of the GLM (e.g. Z-value), however, correspond to the standard representation of the results of a GLM and thus, we omit to explain them in more detail.

RC2.24) P8, L21: Where are the results illustrating the influence of the forest structure? The only results presented are simulation scenarios with random or no forest (Fig4 and Fig5).

We decided to show results of two different forest types and no forest in fig. 4 and 5 as we do not want to overload the figure. The influence of the forest structure is reported in the text and can be seen in fig. 9 (intensity-frequency curves) as well as the statistical models.

RC2.25) P9, L1: Why, for the level 8 of the RTFreq model (Fig 8), a volume < 1.1 m3 has a smaller Nrpred value than a volume > 1.1 m3, and why, for the level 11 of the RTFreq model (Fig 8), a volume < 1.2 m3 has a higher Nrpred value than a volume < 1.2 m3? Could you explain the difference?

There was a mistake in the initial graph at level 8: There is a smaller $Nrp_{red}$ value for volume > 1.1 m$^3$ than for volumes < 1.1 m$^3$. The graph was adapted according to the new regression tree model based on simulations for all source cells.

RC2.26) P9, L30: Which data were used to build the Fig 9?

We corrected the description of the figure according to "Specific comments 3, 5."

**Specific comments 5**

RC2.27) P10, L22-32: This paragraph is currently a description of your results. A comparison between your results and other one from the literature would improve significantly the discussion. Here is a list of recent papers working on similar subjects:

We thank for the interesting literature suggestions and integrated part of them in our discussion. However, we did not substantially change the mentioned paragraph as we do not only describe the results but also highlight and summarize their significance. A discussion and comparison with other literature follow in the next paragraph(s).

**Other comments**

RC2.28) P3, L17-20: This two sentences should be located in the material and methods section. In the introduction, it would be appreciated to have a short paragraph describing the different rockfall models that can be found in the literature and highlight the few one that take into account the protection effect of the forest.

We made reference to Volkwein et al. 2011 who summarize and describe in detail a wide variety of existing rockfall models (P4 L19).

RC2.29) P4, L5: A justification for the choice of a concave profile with slope angle varying between 20 to 40° needs to be added.

We chose a concave profile as this corresponds to typical and probably most frequent slope geometries of rockfall slopes. In order to test the influence of the slope profile on the results, we validated the statistical models with results of simulations on linear slopes with slope angles of 32°, 35°, 38° and 40°.

RC2.30) P4, L6: Why do you add random slope angle variation to your profile? Adding this angle variation comes in conflict with the "controlled conditions" you are looking for P4, L4.

RC2.31) P4, L10: Why do you add a road into your slope profile? Adding a road does not appear to be necessary for the analysis done in the paper. Its influence is never

presented in the result section nor in the discussion and conclusion section.

We randomly varied the slope angle of each cell aiming at more realistic conditions. We agree, however, that this is slightly contradictory to the "controlled conditions". For this reason, we decided to run the new simulations without slope angle variation and we also omitted the road in the new slope as it is indeed not necessary for the presented analysis.

RC2.32) P7, L16: Why do you choose to calculate cbA using a slope width of 100 m? Did you test other widths and analyse their influence on this indicator?

The slope width of 100 m is used to normalize the cumulative basal area for a given runout distance. As such, it does not make any difference whether we use 1, 10 or 100 m.

RC2.33) P8: Can you give a ranking of the influence of the different parameters for the 4 statistical models?

The p-values of the parameters and the position in the regression tree can be regarded as ranking. This, however, has to be interpreted carefully, since the multivariate models reflect interactions of the parameters and a ranking does not necessarily make sense.

[revised manuscript text omitted]

---

## Author Response (AR2)

**Responses to Reviewer Comments**

We thank the reviewer for the second revision of our paper and the helpful suggestions. In the following, we respond to the author's comments (blue) and explain our adaptions (marked red in the manuscript below). We further made a few linguistic adaptations which are not explicitly indicated.

**2. Specific comments**

P4, L10: The justification given for the simulations performed on a concave slope (in the response to the reviewers) could be placed in the article.

We integrated the justification for the concave slope in the text.

P4, L18: The first sentence could be placed in the introduction section.

It is true that this sentence would also fit in the introduction section. However, we decided to place it here to specifically set the context of rockfall simulation models.

P5, L13: The explanation given for the block volume range chosen for the simulations (in the response to the reviewers) could be placed in the article.

As suggested, we complemented the explanation for the block volume range chosen.

P5, L27: As I have mentioned in the first review (Other comments: 3.6.7) the author have mistaken the Fig. 2 and Fig. 3.

Corrected as suggested.

P8, L19/ P3, L19/ P10, L21/ P10, L26: The expression "reducing effect of forest" should be improved as it was suggested in the first review. It could be replaced by "the protection effect of the forest".

We replaced the expression by "protection effect of forest" or "reduction by the forest".

Tab 3: The variable (Intercept) remain undefined.

We added the intercept to the table description.

Fig 8: In the Rt_int model the extreme values observed around -1.5 E90_red have to be explained or discussed. In addition, due to this extreme value, the scale of the box plots could be improve.

We explained this with the following answer (P9 L25): *These cases represent blocks passing the lowest evaluation line at 480 meters under forested conditions, which have relatively high energies compared to non-forested conditions ($E90_F$ = 118.1 kJ; $E90_{nF}$ = 47.1 kJ). Only these few higher energy blocks (e.g. $Nrp_F$ = 3; $Nrp_{nF}$ = 398) are able to reach such runout distances under forested conditions and strongly determine the statistics.*

We agree with your comment regarding the scale of the box plots but did not find a satisfying solution. When changing the scale of box, the extreme values will disappear. Therefore, we decided to maintain the scale as it is.

Fig 9: New symbols need to be chosen to differentiate the curves aisle, clustered and gaps.

We changed the symbols in the plot.

P10, L5: As I have already noticed in the first review the reference cited is not accurate. The result presented in Lopez et al 2016 is : "143 years for forested condition in 1850 to > 2000 years for forested condition in 2013 for a block volume of 1.2 m3". There was less forest in 1850 compared to 2013 but there was forest.

We adapted this in the text.

P10, L22: The word elasticity can't be used to describe the soil response during a block impact. It could be replaced by "capacity of the soil to dissipate energy". See the specific comments 3.1.6 of the first review.

Changed as suggested.

[revised manuscript text omitted]